



# The Significance of the Leaf-Area-Index on the Evapotranspiration Estimation in SWAT-T for Characteristic Land Cover Types of Western Africa

Fabian Merk[1], Timo Schaffhauser[1], Faizan Anwar[1], Ye Tuo[1], Jean-Martial Cohard[2], and Markus Disse[1]

[1]School of Engineering and Design, Technical University of Munich, Munich. Germany
[2]Institute of Engineering and Management, University of Grenoble Alpes, Grenobles, France

**Correspondence:** Fabian Merk (fabian.merk@tum.de)

**Abstract.**

Evapotranspiration (ET) plays a pivotal role in the terrestrial water cycle in sub-humid and tropical regions. Thereby, the contribution of plant transpiration can be distinctively greater than the soil evaporation. The seasonal dynamics of plant phenology, e.g., commonly represented as the vegetation attribute leaf-area-index (LAI), closely correlates with actual ET (AET).

Addressing the reciprocal LAI-AET interaction is hence essential for practitioners and researchers to comprehensively quantify the hydrological processes in water resources management, particularly in the perennially vegetated regions of Western Africa. However due to the lack of field measurements, the evaluation of the LAI-AET interaction still remains challenging. Hence, our study aims to improve the understanding of the role of LAI on the AET estimation with the investigation of characteristic regions of Western Africa. We setup eco-hydrological models (SWAT-T) for two homogeneous land cover types (forest and

grassland) to guarantee the representativeness of field measurements for LAI and AET. To evaluate the LAI-AET interaction in SWAT-T, we apply different potential ET methods (Hargreaves, Penman-Monteith (PET-PM), Priestley-Taylor). Further, the parameter sensitivity for 27 relevant LAI-AET parameters is quantified with the elementary effects method. The comprehensive parameter set is then optimized using the Shuffled-Complex-Evolution algorithm. Finally, we apply a benchmark test to assess the performance of SWAT-T to simulate AET and to determine the relevance of a detailed LAI modelling. The results

show that SWAT-T is capable to accurately predict LAI and AET on the footprint scale. While all three PET methods facilitate an adequate modelling of LAI and AET, PET-PM outperforms the methods for AET independent of the land cover type. Moreover, the benchmarking highlights that if an optimization process only accounts for LAI but disregards AET data, its prediction of AET still yields an adequate performance with SWAT-T for all PET methods and land cover types. Our findings demonstrate that the significance of a detailed LAI modelling on the AET estimation is more pronounced in the forested than

in the grassland region.

## 1   Introduction

Evapotranspiration (ET) is a key hydrological process of the continental water cycle, particularly in the sub-humid and tropical regions of Western Africa where the share of ET to precipitation can be up to 70 – 80 % (Rodell et al., 2015). The high share of





ET in the water cycle inevitably necessitates the reliable estimation of ET for water resources studies on all scales in sub-humid and tropical regions. Concurrently, the accurate computation of ET remains challenging for researchers and practitioners since ET is dynamic in space and time (Michel et al., 2016; Miralles et al., 2016). It varies notably dependent on land cover, soil properties, water availability, vegetation state, and time of the year, even in proximate regions (Chu et al., 2021). In addition, plant transpiration has a decisive contribution to the total evapotranspiration (Gerten et al., 2004; Schlesinger and Jasechko, 2014; Miralles et al., 2016; Wei et al., 2017). It is directly linked to the canopy conductance which strongly correlates with the leaf-area-index (LAI) (Good et al., 2014; Wang et al., 2014). Thus in perennially vegetated regions with high transpiration rates, such as the sub-humid Western Africa, the LAI-ET interaction plays a pivotal role in the ET quantification (Schlesinger and Jasechko, 2014; Wei et al., 2017; Bright et al., 2022).

Albeit its importance, the availability of LAI and ET ground measurements is scarce. In previous ET studies, authors have used existing global monitoring networks, such as eddy covariance (EC) systems (e.g., AmeriFlux (Novick et al., 2018), AMMA-CATCH (Galle et al., 2018), or FLUXNET (Friend et al., 2007)), to inform catchment-scale hydrological models to comprehensively assess all processes of the hydrological cycle (Schneider et al., 2007; Hector et al., 2018; Ferreira et al., 2021; Jepsen et al., 2021; López-Ramírez et al., 2021). Still, the derived AET estimates from EC systems can not be extrapolated without limitations beyond the location site. The usually small source area that contributes to the measured fluxes is defined as flux footprint. Dependent on soil and land cover properties underlying the footprints, the source area spatially limits the representativeness of the AET measurements (Chu et al., 2021). For LAI, the limited availability of field observations is commonly addressed with the exploitation of satellite-based LAI data. A favorable data set is thereby the Global Land Surface Satellite (GLASS) LAI data (Liang et al., 2021) where the widely used MODIS LAI data has been advanced with machine learning applications on the global scale (Liang et al., 2014). The validation reports of GLASS-LAI data present accurate LAI time series results, particularly in perennially vegetated regions (Liang et al., 2014) where the satellite based vegetation data can be subject to noise and cloud influences (Viovy et al., 1992; Strauch and Volk, 2013; Atkinson et al., 2012; Alemayehu et al., 2017).

In the present study, the semi-distributed, physically-based eco-hydrological Soil and Water Assessment Tool for the tropics (SWAT-T) (Alemayehu et al., 2017) is applied. The SWAT-T model is a modification of SWAT (Arnold et al., 1998) which has been introduced by Strauch and Volk (2013) and further developed by Alemayehu et al. (2017) to account for a more realistic plant growth modelling of perennial vegetation in tropical regions. The merits of SWAT-T for an improved prediction of LAI and AET have been highlighted in different tropical and sub-humid regions. It has been applied on the catchment scale in Eastern Africa (Alemayehu et al., 2017), in Brazil (Ferreira et al., 2021) as well as in Australia (Zhang et al., 2020) and on the micro-catchment scale in Mexico (López-Ramírez et al., 2021). Moreover, the application of SWAT-T for climate impact assessment has been presented in Peru on the catchment scale (Fernandez-Palomino et al., 2021). Remotely-sensed AET has been mostly employed to assess the model fitness of simulated AET (Alemayehu et al., 2017; Zhang et al., 2020; Fernandez-Palomino et al., 2021; Ferreira et al., 2021). For the African continent, remotely-sensed AET products can however be limited due to uncertainties in their reliability (Weerasinghe et al., 2020).



For the LAI estimation, the aforementioned SWAT-T studies relied on the application of remotely-sensed LAI from MODIS. In the past, measured LAI was used with SWAT (Park et al., 2017; Yang et al., 2018; Nantasaksiri et al., 2021) as well as

observed forest biomass production (Khanal and Parajuli, 2014) for an analysis of the LAI model parameters. The number of parameters thereby differ from three (Yang et al., 2018) to eight (Nantasaksiri et al., 2021) parameters. Alemayehu et al. (2017) suggests the calibration of 11 LAI parameters. LAI and AET are correlated and influence each other in SWAT/SWAT-T (Arnold et al., 1998). For example, the water stress on plants is dependent on AET and can determine the actual plant growth in SWAT/SWAT-T (Neitsch et al., 2011). When modelling LAI, a consideration of the relevant AET parameters is hence essential.

To the best of our knowledge, the reciprocal LAI-AET interaction and the relevance of a coupled LAI-AET parameter estimation have not yet been evaluated with respect to measured LAI and AET in SWAT/SWAT-T. The influence of LAI on AET in the application of SWAT/SWAT-T has covered either not all relevant LAI-AET parameters, has been studied for heterogeneous source areas for AET, or has used only remotely-sensed AET and LAI data. Hence, we address these shortcomings and focus our research on the comprehensive evaluation of the significance of LAI on AET in SWAT-T. Further, we test the hypothesis if

a detailed plant growth model optimization (single LAI optimization regarding observed or GLASS-LAI) can still adequately estimate AET with SWAT-T.

We evaluate the LAI-AET interaction for two typical, perennially vegetated land cover types of Western Africa using a SWAT-T model on the seamless footprint scale of the EC system for each site. The sites are located in the sub-humid Bétérou catchment in Benin. First, we highlight the relevance of a coupled LAI-AET parameter estimation for the prediction of LAI.

Then, a global sensitivity analysis with the elementary effects methods (Morris, 1991) is applied to quantify the parameter sensitivities and to enable a ranking of the sensitivity levels. We optimize the LAI-AET parameters with LAI data (observed and GLASS-LAI), exclude AET as a proxy in the model optimization, and eventually evaluate the AET model response of the LAI optimization. For this purpose, the performance test proposed by Seibert et al. (2018) is conducted. The test compares the best optimized model (simultaneous LAI and AET optimization as upper benchmark) and the single LAI optimization approaches

(observed or GLASS-LAI). To provide a lower limit of the general LAI-AET performance of SWAT-T, a random sampling approach of the LAI-AET parameters (lower benchmark) is applied. The LAI-AET parameter optimization is conducted using the Shuffled-Complex-Algorithm (SCE-UA) (Duan et al., 1994).

## 2 Methods

Fig. 1 gives an overview of the methods applied in this study to evaluate the significance of LAI on the AET estimation in

SWAT-T. At first, the input data is processed and footprint scales SWAT-T models for two characteristic, perennially vegetated regions in Western Africa are setup. The relevance of a coupled LAI-AET parameter estimation is investigated with one-at-a-time parameter changes and evaluated regarded observed LAI and AET data. The sensitivity analysis is conducted based on the elementary effects method with respect to observed LAI. Finally, the role of LAI on the AET estimation in SWAT-T is assessed with an optimization approach (Shuffled-Complex-Evolution algorithm) and the performance to predict AET is tested with a

benchmark test. In the following, the steps of the methodology are explained in detail.



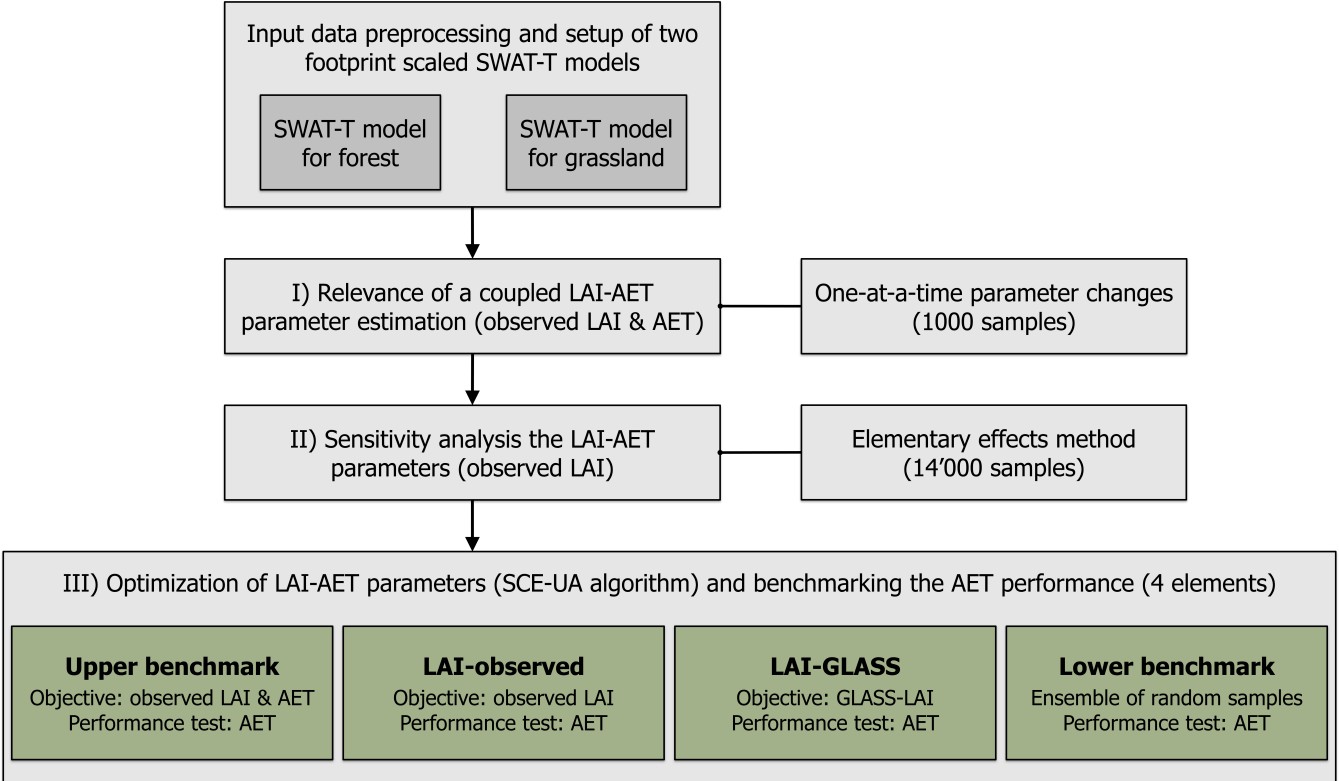

**Figure 1.** Methods to assess the significance of LAI on AET estimation as applied in the present study. For all three steps of the methodology, three different PET methods (Hargreaves, Priestley-Taylor, Penman-Monteith) available in SWAT-T are applied.

## 2.1 Model description and parameter selection

The SWAT-T model is a further development of the ecohydrological model SWAT (Arnold et al., 1998). In SWAT-T, the plant growth module has been modified to account for a more realistic perennial plant phenology in tropical regions (Alemayehu et al., 2017) which can eventually improve the AET prediction (Zhang et al., 2020; Fernandez-Palomino et al., 2021; Ferreira et al., 2021; López-Ramírez et al., 2021). Apart from the plant growth module, SWAT-T and SWAT are identical. The original SWAT model has been applied worldwide in different river basins (Arnold and Fohrer, 2005; Tan et al., 2020) as well as regionally in Benin (Akoko et al., 2021). Specifically in Benin, most of the applications focused on the discharge assessment for the Ouémé river basin (Bossa et al., 2014; Poméon et al., 2018) and its tributaries (Giertz et al., 2006; Bossa et al., 2012; Duku et al., 2016, 2018; Danvi et al., 2017; Togbévi et al., 2020). In previous studies in Western Africa, remotely-sensed AET was also used as a main calibration objective to predict streamflow (Odusanya et al., 2019, 2021). To the best of our knowledge, the SWAT-T model has so far not been applied in Benin but in Eastern Africa (Alemayehu et al., 2017).

Generally, the SWAT/SWAT-T model is spatially discretized into subbasins and further subdivided in hydrological response units (HRUs). Three options are available to compute the potential ET (PET) in SWAT/SWAT-T: the temperature-based Har-



greaves (PET-HG) (Hargreaves and Samani, 1985), energy-based Priestley-Taylor (PET-PT) (Priestley and Taylor, 1972) and
the combined temperature and energy-based Penman-Monteith (PET-PM) (Monteith, 1965) method. Tab. 1 summarizes the
equations how PET $E_0$ is computed and highlights the integral part of LAI in each approach.

**Table 1.** Approaches to compute potential evapotranspiration $E_0$ and potential transpiration $T_{plant}$ provided in SWAT-T.

| PET method | Equation for $E_0$ | Equation for $T_{plant}$ |
|---|---|---|
| PET-HG | $E_0 = \dfrac{0.0023 \cdot H_0}{\lambda} \cdot \sqrt{T_{mx} - T_{mn}} \cdot (T_{av} + 17.8)$ | $T_{plant} = \begin{cases} LAI \cdot \frac{E_0}{3.0}, & \text{if } LAI \leq 3.0 \\ E_0, & \text{if } LAI > 3.0 \end{cases}$ |
| PET-PT | $E_0 = \dfrac{\alpha_{pet} \cdot \Delta}{\lambda \cdot (\Delta + \gamma)} \cdot (H_{net} - G)$ | $T_{plant} = \begin{cases} LAI \cdot \frac{E_0}{3.0}, & \text{if } LAI \leq 3.0 \\ E_0, & \text{if } LAI > 3.0 \end{cases}$ |
| PET-PM | $E_0 = \dfrac{\Delta \cdot (H_{net} - G) + \rho_{air} \cdot c_p \cdot (e_z^0 - e_z)/r_a}{\lambda \cdot (\Delta + \gamma \cdot (1 + r_c/r_a))},$ | $T_{plant} = \dfrac{\Delta \cdot (H_{net} - G) + \rho_{air} \cdot c_p \cdot (e_z^0 - e_z)/r_a}{\lambda \cdot (\Delta + \gamma \cdot (1 + r_c/r_a))},$ |
| | with $r_c$, $r_a$ from alfalfa crop reference | with $r_c$, $r_a$ from actual plant (canopy height and LAI) |

The equations in Tab. 1 are determined in which $\lambda$ is the latent heat of vaporization, $H_0$ is the extraterrestrial radiation,
$T_{mx}$, $T_{mn}$, and $T_{av}$ are the maximal, minimal, and mean daily temperature, respectively, $\alpha_{pet}$ is a coefficient, $\Delta$ is the slope
of the saturation vapor pressure-temperature, $\gamma$ is the psychrometric constant, $H_{net}$ is the net radiation, $G$ is the heat flux
density to the ground, $\rho_{air}$ is the air density, $c_p$ is the specific heat at constant pressure, $e_z^0$ is the saturation vapor pressure of
air at height $z$, $e_z$ is the water vapor pressure of air at height $z$, $r_a$ is the aerodynamic resistance, and $r_c$ is the plant canopy
resistance. In PET-PM, $r_a$ and $r_c$ are attributed to the alfalfa crop reference for the computation of $E_0$ (Neitsch et al., 2011).
After the calculation of $E_0$, it is eventually partitioned in potential plant transpiration $T_{plant}$ and soil evaporation $E_{soil}$. $T_{plant}$
is thereby computed in dependency of the value of LAI for the given day for all PET methods. For PET-HG and PET-PT, a
threshold of $LAI = 3.0$ determines if $T_{plant}$ is equal to $E_0$, i.e., all potential evapotranspiration is coming only from the plant
transpiration without the consideration of soil evaporation. If $LAI \leq 3.0$, a share of $E_0$ is potentially available for $T_{plant}$ and
$E_{soil}$. For PET-PM, $T_{plant}$ is computed using the equation of Penman-Monteith (Tab. 1) where $r_a$ and $r_c$ are determined with
respect to the actually modelled plant canopy and LAI. $E_{soil}$ for PET-PM is then $E_{soil} = E_0 - T_{plant}$. Eventually, the actual
plant transpiration and soil evaporation are computed dependent on the water availability and different biophysical parameters,
such as LAI or root depth, as well as soil properties, such as the field capacity. Actual plant transpiration and soil evaporation
are then summed to the actual ET (AET).



Plant growth is modelled in SWAT/SWAT-T with the "Environmental Policy Impact Climate" (EPIC) model (Arnold et al., 1998) where LAI is a key vegetation attribute for the plant phenology (Neitsch et al., 2011). Generally, the plant growth in SWAT/SWAT-T can be divided in an initial phase (start of the growing phase), a growing phase, a period of maturity (growing is halted to a constant LAI), the leaf senescence phase (natural decline of the plant and a decreasing LAI), and a dormancy period (no plant growth but constant LAI). In the growing phase, the optimal leaf development in SWAT/SWAT-T is computed with:

$$fr_{LAI_{mx}} = \frac{fr_{PHU}}{fr_{PHU} + exp(l_1 - l_2 \cdot fr_{PHU})}, \tag{1}$$

where $fr_{LAI_{mx}}$ is the fraction of the maximum leaf area index of a plant with respect to the fraction of the potential heat units for the plant, $fr_{PHU}$ is the fraction of the potential heat units in the current day of the growth cycle, and $l_1$ and $l_2$ are shape coefficients. The plant growth continues until the maximum leaf area index is reached:

$$\Delta LAI_i = (fr_{LAI_{mx},i} - fr_{LAI_{mx},i-1}) \cdot LAI_{mx} \cdot (1 - exp(5 \cdot (LAI_{i-1} - LAI_{mx}))), \tag{2}$$

For perennial plants, the LAI for the given day $i$ under optimal conditions is eventually computed as:

$$LAI_i = LAI_{i-1} + \Delta LAI_i. \tag{3}$$

However, the optimal plant growth can be constrained in SWAT/SWAT-T due to water, temperature, nitrogen, or phosphorous stress. The water stress $wstrs$ is thereby directly linked to the actual plant transpiration and the total water plant uptake. The temperature stress $tstrs$ is computed based on the air temperature of the given day and the user defined parameters $T_{base}$ and $T_{opt}$. Nitrogen and phosphorus stresses, $nstrs$ and $pstrs$ respectively, are computed to account for insufficient nutrients (see Appendix A2 for the supplementary equations). The actual plant growth is determined with a plant growth factor $\gamma_{reg}$:

$$\gamma_{reg} = 1 - max(wstrs, tstrs, nstrs, pstrs), \tag{4}$$

and the actual leaf area added on a day $i$ is computed as:

$$\Delta LAI_{act,i} = \Delta LAI_i \cdot \gamma_{reg}. \tag{5}$$

The computation of LAI and AET in SWAT/SWAT-T can also be influenced by changes of carbon dioxide levels in the atmosphere (Neitsch et al., 2011). In SWAT/SWAT-T, the leaf conductance is computed based on carbon dioxide concentration for PET-PM. Concurrently, the carbon dioxide concentration is employed in the calculation of the biological efficiency (BIO_E) and impacts the radiation use efficiency of the plants. If different $CO_2$ levels are simulated, substantial changes in LAI and AET can be observed (Ficklin et al., 2009; Wu et al., 2012). In the present study, the carbon dioxide level is constant but variations should be further investigated in the future.

The major difference between the plant growth modelling in SWAT and SWAT-T are two features: a logarithmic decline of LAI and the automatic start of the growing phase based on a soil moisture index. In the first plant growth modification





of SWAT, Strauch and Volk (2013) introduced a logarithmic decline of LAI in the leaf senescence phase for a more realistic representation of the LAI decrease and to avoid a rapid drop of LAI:

$$LAI = \frac{LAI_{max} - LAI_{min}}{1 + exp(-t)}, \tag{6}$$

where t is defined considering the fraction of the potential heat units at which senescence become the dominant growth phase

$fr_{PHU_{sen}}$ as:

$$t = 12 \cdot \left( \frac{1 - fr_{PHU}}{1 - fr_{PHU_{sen}}} - 0.5 \right) if \ fr_{PHU} \geq fr_{PHU_{sen}}. \tag{7}$$

Since the plant growth in the tropics is generally governed by the water availability in the soils (Jolly and Running, 2004), Alemayehu et al. (2017) further modified the SWAT version of Strauch and Volk (2013) and implemented an automatic start of the growing phase which is triggered by the soil moisture index. For this purpose, the soil moisture index $SMI = P/E_0$

is introduced. The precipitation $P$ is aggregated for a user defined time window (here: 5 days). A $SMI$ threshold to start the growing has to be defined (here: $SMI = 0.5$). To avoid false starts of the new growing cycle, the end of the dry season (SOS$_1$, here: October) and the beginning of the rainy season (SOS$_2$, here: January) have to be specified by the user, too (Alemayehu et al., 2017). In SWAT, the start of the growing phase is linked to the number of accumulated heat units. In SWAT-T, the soil moisture index has replaced this dependency of the heat units. The heat units are mainly used in SWAT-T to define the plant

growth development over the year (see Equ. 1).

27 parameters have been selected to investigate the LAI-AET interaction (Tab. 2). The selection of LAI parameters follows the suggestion of Alemayehu et al. (2017), whereas the AET parameters are chosen based on literature review. In the past, 27 parameters of SWAT have been assessed for sensitivity analysis with particular focus on AET (Ha et al., 2018; Odusanya et al., 2019; Bennour et al., 2022; Koltsida and Kallioras, 2022). Parameters with a coinciding low sensitivity reported in these studies,

e.g., the hydraulic conductivity in the channel (CH_K2) or groundwater baseflow delay (GW_DELAY), are not considered in the present LAI-AET parameter list to reduce the total parameter space. For the present study, the soil layer thickness (SOL_D) is given for four soil layers from field measurements (Judex and Thamm, 2008). We adjust only the depth of the lowest soil layer to not excessively shape the ground-truth observations, but to still facilitate an evaluation of the influence the total soil thickness implies on the LAI-AET interaction in SWAT-T.

## 2.2 Study site and footprint-scaled models

The study sites Bellefoungou and Naholou are located in the Western part of the Bétérou catchment (Fig. 2). The climate is typical for Sub-Saharan, sub-humid Africa. The annual precipitation can range from 1100 to 1500 mm (Mamadou et al., 2016; Bliefernicht et al., 2019). The precipitation pattern is unimodal with a rainy season between April and October wheareas from November to March the dry season is occuring. The annual mean daily temperature is 25°C (Galle et al., 2018). The soils in

the Bétérou catchment consist of ferric soils with loamy sand present in the upper soil horizons (Giertz and Diekkrüger, 2003).

The forested Bellefoungou region (latitude 9.791°N, longitude 1.718°E, 445 masl) is covered with typical, widespread woodland ("clear forest") for Sub-Saharan Africa (Ago et al., 2016). The Naholou region (latitude 9.74°N, longitude 1.60°E,





**Table 2.** List of parameters used to estimate LAI and AET with their description.

| Parameter | Description [unit] |
| --- | --- |
| *Parameters associated with plant growth (LAI) in the plant data base of SWAT* | |
| BIO_E | Radiation-use efficiency [(kg/ha)/(MJ/m$^2$)] |
| BLAI | Maximum potential leaf area index [m$^2$/m$^2$] |
| FRGRW$_1$ | Fraction of PHU corresponding to the first point on the optimal leaf area development curve [-] |
| LAIMX$_1$ | Fraction of BLAI corresponding to the first point on the optimal leaf area development curve [-] |
| FRGRW$_2$ | Fraction of PHU corresponding to the second point on the optimal leaf area development curve [-] |
| LAIMX$_2$ | Fraction of BLAI corresponding to the second point on the optimal leaf area development curve [-] |
| DLAI | Fraction of total PHU when leaf area begins to decline [-] |
| T_OPT | Optimal temperature for plant growth [°C] |
| T_BASE | Minimum temperature for plant growth [°C] |
| ALAI_MIN | Minimum leaf area index for plant during dormant period [m$^2$/m$^2$] |
| PHU | Total number of heat units needed to bring plant to maturity [-] |
| GSI | Maximum stomatal conductance [m/s] |
| *Parameters associated with AET estimation* | |
| CAN_MX | Maximum canopy storage [mm] |
| ESCO | Soil evaporation compensation factor [-] |
| EPCO | Plant uptake compensation factor [-] |
| HRU_SLP | Average slope steepness [m/m] |
| SLSUBBSN | Average slope length [m] |
| CN2 | Initial SCS runoff curve number [-] |
| SOL_AWC | Available water capacity of the soil layer [mm] |
| SOL_BD | Moist bulk density [g/cm$^3$] |
| SOL_CBN | Organic carbon content [% soil weight] |
| SOL_K | Saturated hydraulic conductivity [mm/hr] |
| SOL_RD | Maximum rooting depth of soil profile [mm] |
| SOL_D[a] | Soil layer depth [mm] |
| GW_REVAP | Groundwater re-evaporation coefficient [-] |
| RCHRG_DP | Deep aquifer percolation fraction [-] |
| REVAPMN | Threshold depth of water for re-evaporation to occur [mm] |

[a]Here: lowest soil layer depth

449 masl) is largely covered by characteristic mixture of crops and savannah grassland and fallows (Ago et al., 2014). Due to the high share of grassland, the Naholou region is hence defined as grassland region in the following. The estimated flux

footprint extent for the grassland region is 4000 m$^2$, while for the forested region it is seasonally varying and can be up to 60000 m$^2$ (Mamadou et al., 2014). AET accounts for high shares of precipitation at both sites where the share of annual AET

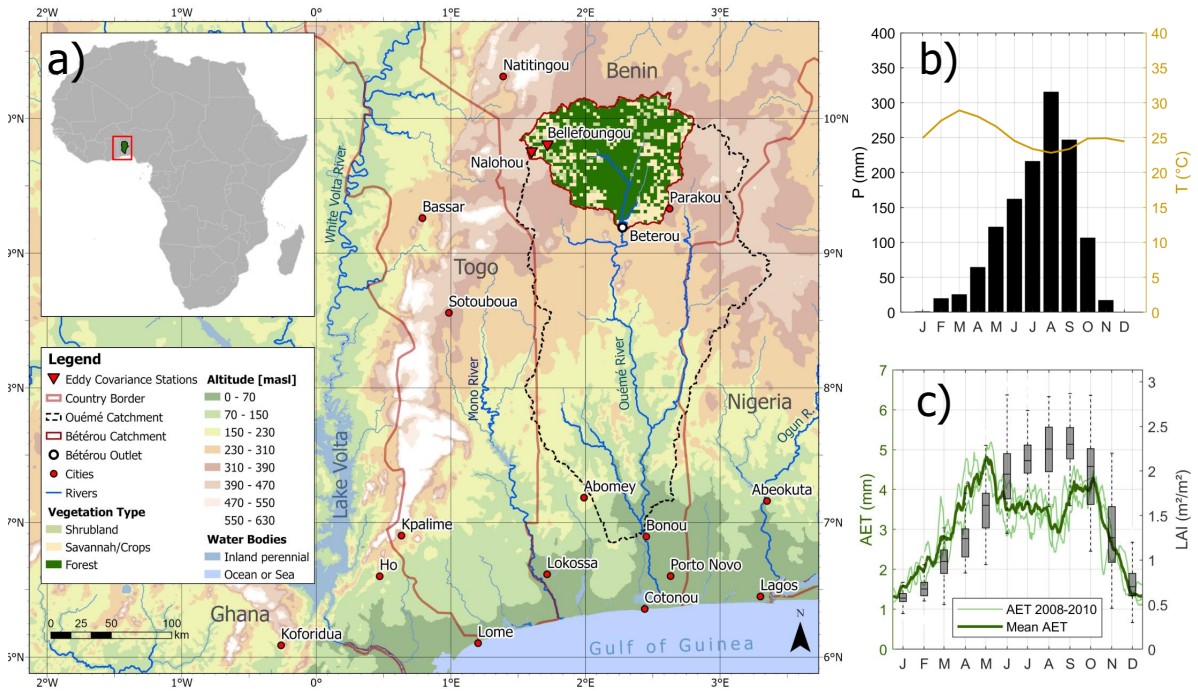

**Figure 2.** (a) Overview of the Bétérou catchment and the locations of the eddy covariance systems (Naholou and Bellefoungou); (b) the seasonality of precipitation and temperature in Bellefoungou; (c) and the correlation of observed AET (in green) and satellite-based LAI (in gray) in Bellefoungou.

to precipitation is 57 % and 72 % for Naholou and Bellefoungou, respectively (Mamadou et al., 2016). The evaporative fraction (EF) of plant transpiration compared to soil evaporation is partiuclarly high in the wet season. The share of plant transpiration to soil evaporation with values of 70 ± 2.5 % at Naholou and 75 ± 0.7 % at Bellefoungou underline the significance of plant

growth to AET in these regions (Mamadou et al., 2016; Hector et al., 2018). The field measurements for LAI in both sites were delineated from hemispherical photographs and the processing methodology proposed by Weiss et al. (2004). The in situ data is complemented with corrections of an ensemble of satellite-based LAI products (CYCLOPE, MODIS, SEVIRI) (Mamadou et al., 2014).

   One SWAT-T model is setup for the forested and the grassland site in the Bétérou catchment. No footprint calculations
for the sites nor the necessary data to compute those is available. We thus adhere the suggestion of Chu et al. (2021) where radii of $< 250\ m$ around flux towers assure flux representativness. The SWAT/SWAT-T models are watershed models. For the model delineation with SWAT-T, we drew circles (250 m radius) around each flux tower to guarantee the representativness of the flux footprints. Based on the underlying DEM, the resulting watershed extents are 8500 m$^2$ and 2300 m$^2$ for the forested and grassland site, respectively. Although the footprint extent in the forested region can be larger depending on the season
(Mamadou et al., 2016), we applied a constant extent following the suggestions of Chu et al. (2021). To ensure the homogeneity of land cover and soil properties, each SWAT-T model consists of a single HRU. LAI and AET are simulated on the daily time





**Table 3.** Overview of the data sets that are applied in this study.

| Variable | Resolution | Database name or source |
|---|---|---|
| Digital elevation model | Rastered DEM, 30 m x 30 m | Copernicus GLO-30 (Copernicus, 2022) |
| Soil map | Soil type clusters | IMPETUS Soil Map (Judex and Thamm, 2008) |
| Observed AET | Daily, pointwise | Mamadou et al. (2016) |
| Observed LAI | Daily, pointwise | Ago et al. (2014); Mamadou et al. (2016) |
| GLASS-LAI | Rastered, 250 m x 250 m | GLASS-LAI (Liang et al., 2021) |
| Precipitation | Daily, pointwise | AMMA-Catch network (Galle et al., 2018) |
| Temperature | Daily, pointwise | AMMA-Catch network (Galle et al., 2018) |
| Solar radiation | Daily, pointwise | AMMA-Catch network (Galle et al., 2018) |
| Relative humidity | Daily, pointwise | AMMA-Catch network (Galle et al., 2018) |
| Wind speed | Daily, pointwise | AMMA-Catch network (Galle et al., 2018) |

step. The data sets used in this study are listed in Tab. 3. The land cover type for each site (forest, grassland) represented in the model is provided in Ago et al. (2014, 2016). We defined the land uses classes "FRSD" and "RNGE" from the SWAT crop data base to the forested and grassland region, respectively. The observed AET data for both sites is available from 1.1.2008 to 31.12.2010 (Mamadou et al., 2016). The observed LAI data for the forested region (Bellefoungou) is available from 1.7.2008 to 31.5.2010 (Ago et al., 2016), for the grassland region (Naholou) from 5.8.2007 to 2.1.2010 (Ago et al., 2014). The meteorological data provided by the AMMA-Catch network dates from 2005 to 2020 (Galle et al., 2018). The GLASS-LAI data is provided from 2000 to 2021 (Liang et al., 2021). To enable the best possible overlap of measured LAI and AET data, the study periods from 1.1.2008 to 31.12.2010 and from 1.1.2007 to 31.12.2010 for the forested and grassland region are defined, respectively.

### 2.3 Evaluation of the coupled LAI-AET parameter estimation

We postulate that for a comprehensive plant growth modelling in SWAT-T both, the LAI and AET model parameters are decisive, particularly if the accurate estimation of plant transpiration is a modelling objective. To assess the relevance of a coupled LAI-AET parameter estimation, we apply parameter changes and compare the corresponding model responses to observed AET and LAI. Each parameter from Tab. 2 is randomly sampled (1000 samples) and the model is run individually for any sample. In each simulation, the other model parameters remain unaltered (one-at-a-time parameter changes). To avoid influences from poorly estimated parameter values (e.g., the default settings), the optimized model parameters from the LAI-AET optimization (cf. Section 2.5) are prescribed for the unaltered parameters. The model response of a parameter change is evaluated in two ways: with respect to observed AET and to observed LAI. Eventually, an evaluation of how LAI parameters influence the AET responses as well as how AET parameters influence the LAI responses is presented with this approach. The analysis of the LAI-AET parameter estimation is conducted for three different PET methods (Hargreaves, Penman-Monteith, Priestley-Taylor) for the forested study site.





## 2.4 Sensitivity analysis with the Morris method

To address the parameter-response-complexity of the coupled LAI-AET modelling with SWAT-T, a sensitivity analysis for all
225 LAI-AET parameters is conducted. Sensitivity analysis is an essential, yet challenging step in the application of hydrological
models and the evaluation of reliable parameters sets, particularly with respect to model equifinality. Different approaches
exist to quantify the model responses to parameter changes. In this study, we take advantage of the elementary effects method,
or Morris method (Morris, 1991), since its computational demand is inexpensive, the parameter sensitivity is statistically
quantified, and non-linear model responses can be determined (Morris, 1991; Campolongo et al., 2007). Moreover, parameters
which are involved in parameter interactions and non-influential parameters can be identified.

Generally, the Morris method screens through a total sample size $N$ where one parameter, or input factor $q = [q_i, ..., q_k]$, is
changed while the others remain constant (one-at-a-time method). The total sample size $N$ is generated based on $r$ defined
levels and $q$ selected parameters such that $N = r(q + 1)$. Based on each sample, the elementary effects $d_i$ is calculated with:

$$d_i(q) = \frac{f(q_i, ..., q_{i-1}, q_i + \Delta, q_{i+1}, ..., q_k) - f(q)}{\Delta} = \frac{f(q + \Delta e_i) - f(q)}{\Delta}, \tag{8}$$

where $\Delta$ represents the parameter step size, $q + \Delta e_i$ denotes the transformed parameter point, $q = [q_i, ..., q_k]$ is any selected
parameter of $N$, and $e_i$ consists of a vector of zeros but one in the $i^{th}$ element. The local sensitivity of parameter $q$ is described
with the value of $d_i(q)$. For the global sensitivity, the statistical moments $\mu_i$ and $\sigma_i$ as mean and standard deviation from the
distribution of the total sample simulation are used (Morris, 1991). We use the absolute mean $\mu^*$ as proposed by Campolongo
et al. (2007) to not disregard non-monotonic model responses because of opposite signs. The statistical moments for each set
$j$ are:

$$\mu_i^*(q) = \frac{1}{r} \sum_{i=1}^{r} |d_i^j(q)|, \tag{9}$$

$$\sigma_i(q) = \sqrt{\frac{1}{r-1} \sum_{i=1}^{r} (d_i^j(q) - \mu_i)^2}. \tag{10}$$

In this study, we quantify the model performance with the Kling-Gupta efficiency (KGE) (see Appendix A2 for the supple-
245 mentary equations). Moreover, we apply the Latin hypercube sampling to guarantee a widespread input space. Using $r = 500$
and $q = 27$ parameters, the total sample size is $N = 14000$.

## 2.5 Coupled LAI-AET parameter optimization and benchmarking

The LAI-AET parameters are first optimized with respect to different objectives which are (i) a multi-objective optimization
with respect to LAI and AET ("upper benchmark"), (ii) an optimization only with respect to observed LAI data ("LAI-obs"),
and (iii) an optimization only with respect to satellite-based GLASS-LAI data ("LAI-GLASS"). The model performance is
then tested based on the benchmark proposed by Seibert et al. (2018) and the different objectives are compared to each other.
We apply the SCE-UA algorithm (Duan et al., 1994) for the optimization of the LAI-AET parameters.





The SCE-UA is a genetic algorithm where first, samples of the parameters are stochastically generated with respect to the lower and upper bounds of the parameter values. The parameters values are then changed to develop the samples to an optimum, i.e., to the optimal value of an objective function. Here, we use KGE to compare the simulated model output with the observed data. In the algorithm application, the initial sample is divided into several sub-samples (complexes) (Duan et al., 1994). In each complex, varying combinations of parameter values are embedded. Each complex is then used to produce offsprings using the downhill simplex procedure (Nelder and Mead, 1965). The probability for a parameter value to be used in the next complex is proportional to its model fitness, i.e., to the objective function. Parameter values of lower fitness are replaced by the new offspring. The main advantage of the SCE-UA agorithm is the application of (i) mutation where new parameter values in the defined parameter spaces can be spontaneously generated and (ii) shuffling where a recombination of the parameter values in new complexes is conducted (Duan et al., 1994).

In this study, different objectives are defined for the LAI-AET parameter optimization (Tab. 4). At first, the LAI-AET parameters are optimized in a multi-objective way with equally weighted respect to observed AET and LAI ("upper benchmark"). This way, the performance potential of LAI-AET to fit both the AET and plant growth is quantified. Further, we also assess if a detailed plant growth optimization can predict AET using the single LAI optimization approaches ("LAI-obs" and "GLASS-LAI"). We use the SPOTPY toolbox (Houska et al., 2015) for the application of the SCE-UA algorithm.

**Table 4.** Summary of benchmark elements, their optimization approach with the corresponding optimization objectives and their evaluation. For the Lower benchmark, the median KGE of the AET performance of all 1000 random samples is used.

| Benchmark element | Optimization approach | Objective(s) | Evaluation for benchmark |
|---|---|---|---|
| Upper benchmark | SCE-UA | Observed LAI & AET | AET |
| LAI-obs | SCE-UA | Observed LAI | AET |
| LAI-GLASS | SCE-UA | GLASS-LAI | AET |
| Lower benchmark | *Random sampling* | - | AET |

All three optimization approaches are compared with each other based on the individual AET evaluation using the benchmarking approach proposed by Seibert et al. (2018). The basic idea of this benchmark is the comparison of a certain modelling framework to an upper and lower benchmark. The upper benchmark is defined to be the best potential model performance (here: "upper benchmark" in Tab. 4). To avoid an arbitrary good or bad model response from a single parameter set, like the default model parameters, Seibert et al. (2018) propose to use random parameter samples. The lower benchmark is then defined to be the median objective fit of the random parameter samples (Seibert et al., 2018). Here, we generate 1000 uniformly distributed LAI-AET parameters samples, evaluate the simulated AET with observed AET, and determine the overall median KGE performance as lower benchmark. With the upper and lower model limits, the AET prediction performance of optimizing the parameters only for LAI can be benchmarked for the footprint scaled models of the forested and grassland region. The three PET methods (PET-HG, PET-PT, and PET-PM) are thereby used to facilitate a comparison of PET methods. With four benchmark elements (Tab. 4), two land cover types, and three PET methods, 24 setups are hence compared to each other to assess the LAI-AET modelling performance of SWAT-T.





## 3 Results

### 3.1 The relevance of a coupled LAI-AET parameter estimation

The influences of the parameter changes on both modelling objectives (LAI and AET) are displayed in Fig. 3. The evaluation shows the distribution which results from each parameter change for AET and LAI. Influences on simulated LAI and AET result for changes of each of the 27 LAI-AET parameters for all three used PET methods. Fig. 3 exemplary shows the parameters (EPCO, SOL_AWC, PHU, ALAI_MIN, DLAI, T_BASE) where the changes on both, LAI and AET, are the most significant regarding PET-PM in the forested region. The results for PET-HG and PET-PT are illustrated in the Appendix A1.

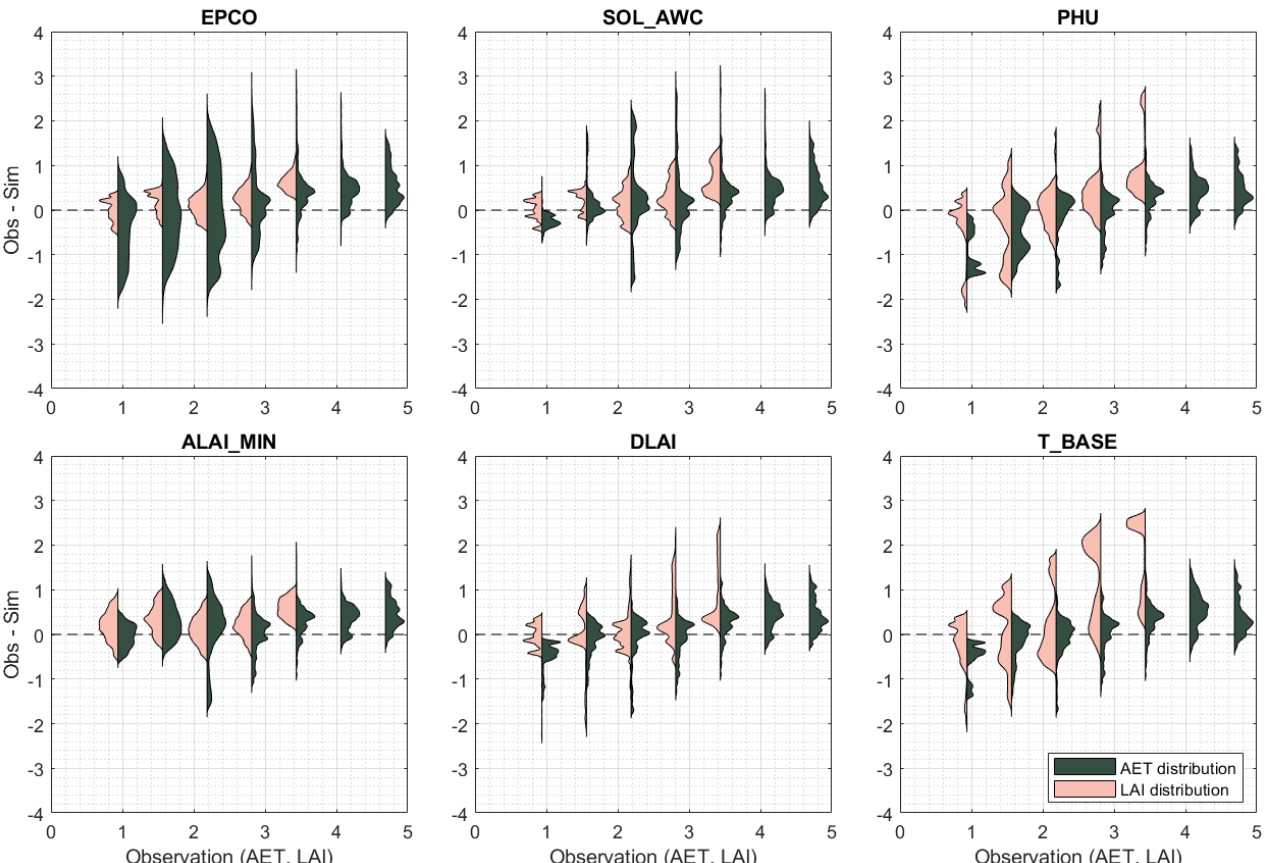

**Figure 3.** Distribution of variations in AET or LAI for the one-at-a-time parameter changes (1000 samples). The distributions are clustered in uniform intervals (size = 0.625) of the observed time series for AET [mm] or LAI [m$^2$/m$^2$]. The x-axis indicates the observed AET [mm] and LAI [m$^2$/m$^2$] values. The y-axis represents the difference between observed and simulated values with $\Delta Y = Obs - Sim$ regarding AET [mm] or LAI [m$^2$/m$^2$]. A perfect fit is indicated with the dashed line for $\Delta Y = 0$. Positive and negative values show an underestimation and overestimation of the simulated values, respectively. The distributions (violin plots) are created based on Karvelis (2024).





As shown in Fig. 3, the variations for the parameter changes regarding intervals of the observed value is displayed. The AET and LAI data is clustered to 7 and 5 intervals, respectively. We used the same interval size for AET and LAI for readability. Each parameter change is then classified according to the interval. Then, the difference between observed and simulated data is calculated ($\Delta Y = Obs - Sim$). The distributions are computed based on the difference within the interval. If for example the observed values for the AET interval of 0.625 to 1.25 mm are compared with the simulated values of the EPCO changes, an overall AET difference of -2.2 to 1.2 mm can be observed. The parameters EPCO and SOL_AWC are commonly associated with AET modelling. Thus, large spreads in the AET model response for the parameter changes can be observed, e.g., the influence of EPCO is particularly high for values AET < 3 mm. Still, influences of both parameters on the LAI simulation are indicated, too. For EPCO, decisive variations in the LAI response are implied for values LAI > 2 m$^2$/m$^2$. As the plant growth is close to the phase of maturity, the significance of the water uptake by the plant in SWAT-T (determined with EPCO) increases and an importance of EPCO for LAI modelling can be observed. Generally, the EPCO parameter governs the actual transpiration which in turn influences the water stress for plants and thus the actual plant growth. The impact of EPCO on LAI is significant especially in the wet season when essential AET rates occur. With high AET, the plant growth stress is intensified in this period. Similarly, the available water capacity in the soil layers (SOL_AWC) influences the LAI response the further the plant is in the growing phase (LAI > 2 m$^2$/m$^2$). Both, the EPCO and SOL_AWC parameter can limit and elevate the plant growth in the wet season.

Concurrently, the shown LAI parameters (PHU, ALAI_MIN, DLAI, T_BASE) can have an influence on both, AET and LAI. The variations of simulated AET regarding LAI parameters changes is particularly significant in the end of the wet and in the dry season (AET < 3 mm). PHU determines when the plant reaches maturity based on the heat unit assumption. Similarly, the DLAI parameter defines when the LAI begins to decline and thus the start of leaf senescence. If the maturity phase is too early or not long enough, the leaf senescence phase starts too early. In these cases for PHU and DLAI, the LAI-AET interaction is impaired and influences on AET can be observed. The ALAI_MIN parameter defines the minimum LAI value for a plant type during the dormant period. If ALAI_MIN is set too small, the plant is underrepresented in the dry period which results in low plant transpiration rates. The parameter changes for T_BASE result in the largest spread of simulated LAI values for all stages of the plant growth phase. With T_BASE, the temperature stress and hence the actual plant growth is determined in SWAT-T. The influence of the T_BASE parameter on AET is present in the wet and dry periods of the AET modelling, too. Notably, the largest spreads of AET based on T_BASE can be observed in for values $AET < 3mm$.

The one-at-a-time parameter change evaluation and the LAI-AET cross-comparison show that AET parameters, such as EPCO or SOL_AWC, can be decisive for the AET and LAI modelling. Fig. 3 also highlights that the LAI parameters, such as PHU, ALAI_MIN, DLAI, or T_BASE, can influence the AET model response. The variations in LAI and AET resulting from changes of the remaining 21 LAI-AET parameters (cf. Tab. 2) are similar, although not shown here. A coupled LAI-AET parameter estimation is hence essential for the reliable computation of LAI and AET, particularly for perennial land cover types in a sub-humid region in Western Africa.





## 3.2 LAI-AET parameter sensitivity analysis with respect to observed LAI

The sensitivities of the LAI-AET parameters are quantified using the elementary effects method regarding the observed LAI data. Fig. 4 shows the statistical moments $\mu^*$ and $\sigma$ for each parameter. It can be observed that nearly all parameters are located close or slightly above than the 1:1 line which defines a non-linearity for the parameters (Garcia Sanchez et al., 2014). Albeit some exceptions, the parameters in the forested region result in higher $\sigma$ values implying that the parameter interactions are more non-linear than in the grassland region. Generally, a proximity of the parameter sensitivities for each land cover type method can be observed, e.g., the diamond symbols for forest are close to each other. Thus, differences for the PET methods and for the same land cover type are not significant which suggests a potential independence of the LAI parameter sensitivity to the PET method. Moreover, all three groundwater parameters result in values $\mu^* = 0$ and are thus insensitive for plant growth. Hence, they are excluded from the in-depth parameter analysis in the following.

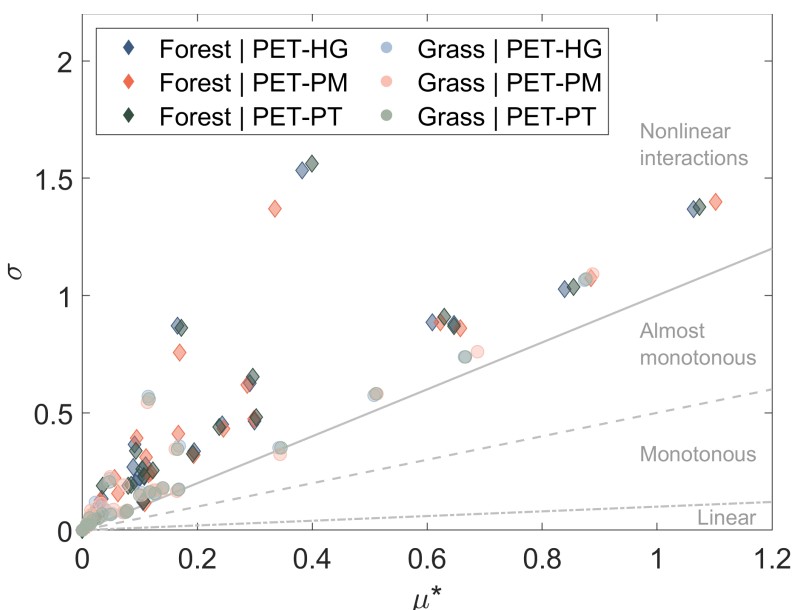

**Figure 4.** The statistical moments $\mu^*$ and $\sigma$ of the elementary effects for the evaluation of the LAI-AET parameter sensitivity with respect to observed LAI. We use the relation of $\sigma/\mu^*$ to classify regions of non-linear, almost monotonous, monotonous and linear parameter behavior (Garcia Sanchez et al., 2014).

Moreover, all PET methods are clustered to compare the sensitivity of the LAI-AET parameters for different land cover types. Fig. 5 shows the distribution of $\mu^*$ where all PET methods are combined in one land cover group. The parameters are ranked according to the mean $\mu^*$ values resulting from the forested region. It can be observed in Fig. 5 that the general parameter sensitivity patterns are similar in the forested and grassland region albeit with differences in the magnitude of $\mu^*$ for the land covers. The most sensitive parameters for both land use types are T_BASE, PHU, DLAI, BLAI, and SOL_RD. Moreover, a high boxchart (high spread of $\mu^*$ values) implies a high parameter interaction. The boxplot heights for the forest





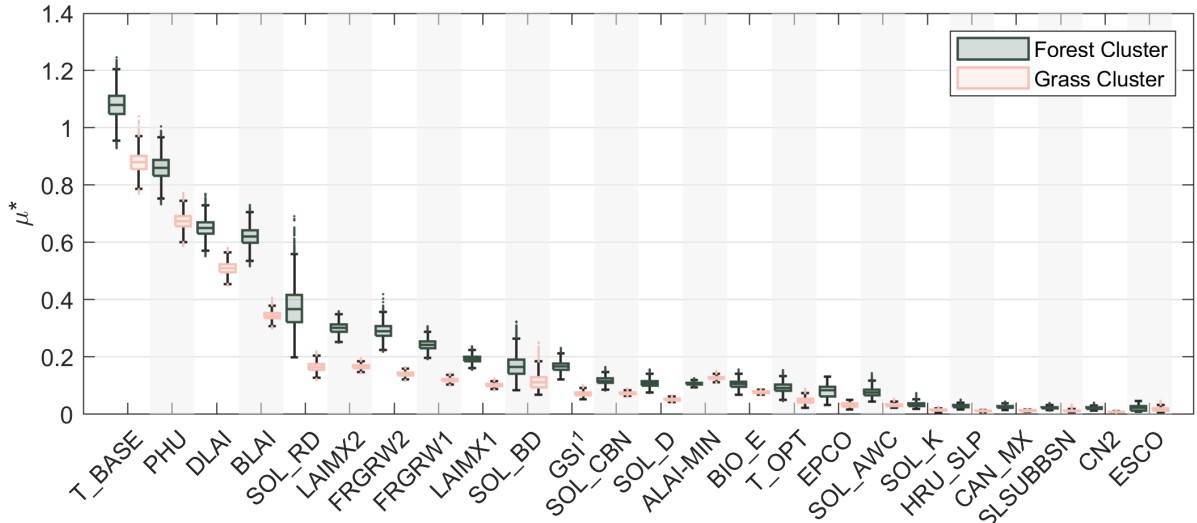

**Figure 5.** Clustering of the sensitivity analysis of all PET methods in the land cover types forest and grassland with respect to observed LAI. The parameters are sorted according to the mean $\mu^*$ values resulting from the forested region. [1]Parameter GSI is only accounted for when PET-PM is used.

and grassland clusters are generally comparable for the same parameters. If a parameter (e.g., T_BASE) in the forest region shows meaningful interactions, alike responses are also indicated for the same parameter in the grassland region. However, the parameters BLAI and SOL_RD appear to have higher parameter interactions in the forest than grassland region. Although the ranking is shown in Fig. 5 with respect to the mean $\mu^*$ values resulting from the forested region, the sensitivity hierarchy of the forested and grassland clusters are generally interchangeable.

From Fig. 5, a clear ranking pattern for PET methods and land use types can be observed (see also Fig. A3). Variations in the ranking position for each parameter are thereby minor, albeit some exceptions for ALAI_MIN, GSI, EPCO, and ESCO. Prior in Fig. 3, the influence of EPCO on LAI has been qualitatively illustrated. Now, the parameter sensitivity of EPCO on LAI is quantified with $\mu^*$ and ranked with the other parameters in Fig. 5. The ranking differs for EPCO, GSI, and ESCO when PET-PM is used (Fig. A3). Its application implies that EPCO and ESCO are less while GSI is more decisive to the LAI model output. The stomatal conductance GSI is only accounted for in SWAT-T when PET-PM is used. Concurrently, the ALAI_MIN parameter is higher ranked for grassland than for forest. Lower LAI values in the dry period of the rainy season increase the parameter ranking of ALAI_MIN. Ultimately, the plant growth parameters are generally higher ranked than the AET parameters. Still, the ranking of SOL_RD, SOL_BD, SOL_CBN, SOL_D, and EPCO indicate an observable influence of AET parameters also on LAI. The sensitivity analysis of the LAI-AET parameters highlights that a coupled LAI-AET parameter estimation is inevitable for a comprehensive assessment of perennial plant growth of SWAT-T in sub-humid regions for all 3 PET methods.





## 3.3 Optimization and benchmark test of the LAI-AET modelling

The SCE-UA algorithm is applied to optimize the LAI-AET parameter in a multi-objective way (upper benchmark) and only
with respect to observed (LAI-obs) as well as satellite-based (LAI-GLASS) LAI data. The evaluation focuses on observed
AET in the following. The upper benchmark optimization results in very good modelling results for three PET methods and
two land cover types.

**Table 5.** Summary of final KGE values with respect to observed AET and LAI for the benchmark elements. For the lower benchmark, the
median AET performance of all 1000 random samples is determined. For LAI modelling, the LAI-GLASS optimization is investigated with
the GLASS-LAI. The lower benchmark LAI values are based on the parametrization of the median AET performance runs.

| PET method | Upper benchmark | | LAI-Obs | | LAI-GLASS | | Lower benchmark | |
|---|---|---|---|---|---|---|---|---|
| | Forest | Grassland | Forest | Grassland | Forest | Grassland | Forest | Grassland |
| *Final KGE values regarding AET performance* | | | | | | | | |
| PET-HG | 0.78 | 0.85 | 0.48 | 0.56 | 0.37 | 0.53 | 0.33 | 0.60 |
| PET-PM | 0.86 | 0.94 | 0.85 | 0.68 | 0.86 | 0.81 | 0.51 | 0.82 |
| PET-PT | 0.77 | 0.88 | 0.55 | 0.66 | 0.50 | 0.65 | 0.47 | 0.69 |
| *Final KGE values regarding LAI performance* | | | | | | | | |
| PET-HG | 0.94 | 0.86 | 0.94 | 0.91 | 0.96 | 0.94 | 0.20 | -0.65 |
| PET-PM | 0.94 | 0.89 | 0.94 | 0.91 | 0.96 | 0.94 | -0.26 | 0.20 |
| PET-PT | 0.90 | 0.87 | 0.94 | 0.91 | 0.96 | 0.94 | -0.36 | -0.76 |

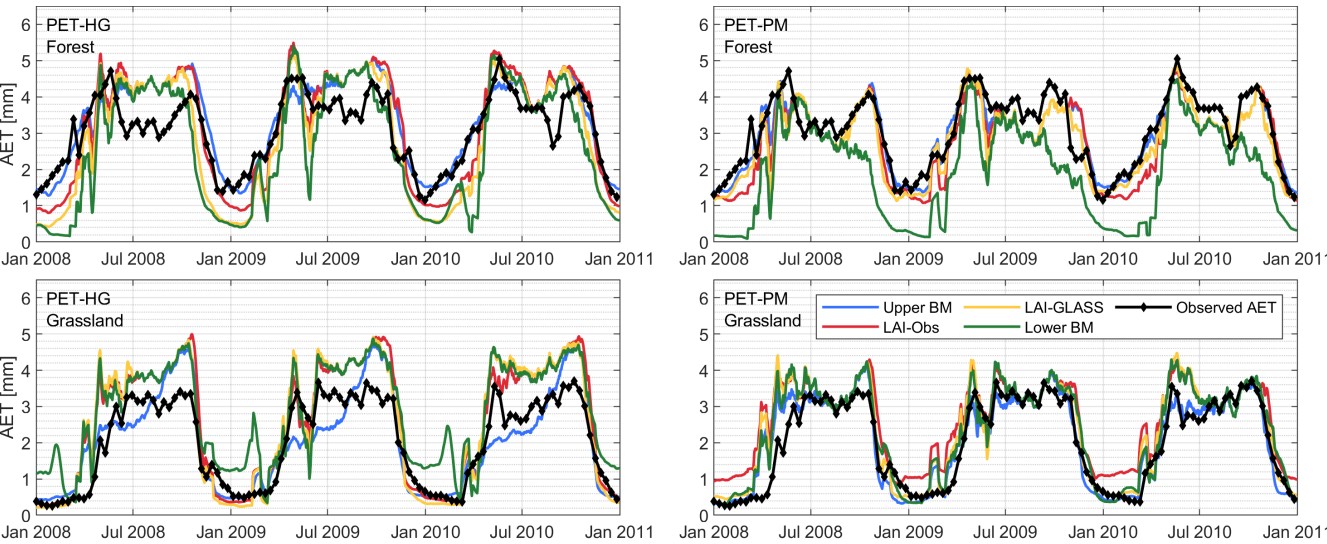

**Figure 6.** Time series of simulated and observed AET for the forested and grassland region. The simulated AET for the four benchmark
elements are displayed for PET-HG and PET-PM.





For all six setups, the model performance (AET) of the upper benchmark is $KGE \geq 0.77$ (Tab. 5). The performance of the LAI optimization to simulate AET results in values of $KGE \geq 0.48$ (LAI-obs) and $KGE \geq 0.37$ (LAI-GLASS). The

median of the random sampling (lower benchmark) determines values of $KGE = 0.33$ to $0.82$ across all six setups for AET. In the forested region, LAI-obs and LAI-GLASS yield better predictions of AET than the lower benchmark. Hence, a single optimization with LAI (observed or GLASS-LAI) can improve the AET estimation in forested regions. If an energy-based PET method (PET-PM or PET-PT) is applied for LAI optimization, an adequate performance for AET can be observed with values of $KGE \geq 0.50$. For the temperature-based PET-HG approach, the AET accuracy is moderate ($KGE = 0.48$, LAI-

Obs; $KGE = 0.37$, LAI-GLASS). In the grassland setups, the lower benchmark outperforms the LAI optimization (observed and GLASS-LAI), although only in small KGE differences. Considering that $KGE \geq 0.5$ is often accepted as a behavioral model performance (Rogelis et al., 2016; Knoben et al., 2019), the resulting KGE values for AET in the grassland setups are still satisfying. For the LAI-obs setups, the variations are $\Delta KGE = 0.04$ for PET-HG, $\Delta KGE = 0.14$ for PET-PM, and $\Delta KGE = 0.03$ for PET-PT. For LAI-GLASS, the KGE differences are $\Delta KGE = 0.07$ for PET-HG, $\Delta KGE = 0.01$ for PET-

PM, and $\Delta KGE = 0.04$ for PET-PT. The main reason for these differences is explained in an overestimation of AET (PET-HG and PET-PM) particularly in the wet period in the grassland region for the LAI optimization (Fig. 6).

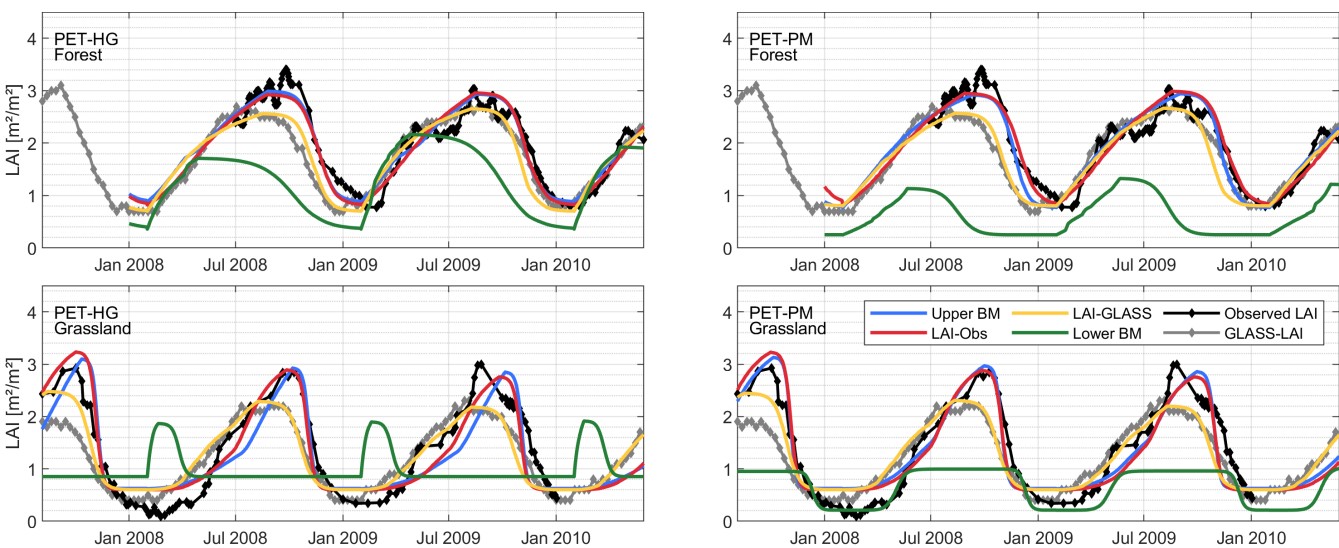

**Figure 7.** Time series of simulated and observed LAI for the forested and grassland region. The simulated LAI for the four benchmark elements are displayed for PET-HG and PET-PM.

Generally, the best model performance is achieved with the application of PET-PM independent of the land cover type. The good AET fit for the LAI optimization approaches is explained with LAI being a term in the calculation of the canopy resistance in the PET-PM equation and the dynamic plant growth cycle. The LAI optimization guarantees a steady transpiration rate even

in the dry period without the plant dying, i.e., LAI dropping to zero. The lower benchmark with no tailor-suited LAI modelling outputs an underestimation of AET in the dry season (Fig. 6) which can be attributed to its low LAI values in this season





(Fig. 7). The simulated LAI and AET data for PET-PT are similar to the PET-HG results (cf. A4). Concurrently, good model performance for PET-PM are achieved for the lower benchmark, too. Although an insufficient LAI modelling ($KGE = -0.26$ for forest, $KGE = 0.20$ for grassland) can be observed, acceptable AET performance is still achieved in the grassland region

(Tab. 5). Ultimately, the results show that SWAT-T is capable to predict accurate LAI and AET predictions. Moreover, the benchmark test shows that even if no AET data is available, the LAI parameter optimization with observed or satellite-based LAI facilitates an acceptable AET estimation in forest and grassland regions. The AET performance from LAI calibration is however constrained to the hydro-meteorological data availability for the choice of the PET method and if the application of energy-based PET methods, particularly PET-PM, is feasible.

## 4   Discussion and outlook

### 4.1   Evaluation of the LAI-AET parameters with observed and GLASS-LAI

In this study, the LAI modelling is evaluated with observed and satellite-based LAI data. Previous studies with SWAT have also employed field measurements for LAI (Park et al., 2017; Yang et al., 2018; Nantasaksiri et al., 2021) or forest biomass production (Khanal and Parajuli, 2014) to evaluated the LAI modelling of SWAT. Yet, the parameters differ in these studies,

e.g., the total number of parameter applied ranges from three (Yang et al., 2018) to eight (Nantasaksiri et al., 2021). The suggested LAI parameter list for SWAT-T in Alemayehu et al. (2017) consists of 11 parameters. We applied one-at-a-time parameter changes to assess the interaction of LAI and AET parameters on both, simulated LAI and AET data. We compared the resulting model responses (LAI, AET) for each parameter change and figured a influences of LAI parameters on AET modelling and vice versa. Although the assessment of the one-at-a-time changes was based on a qualitative analysis, a clear

pattern of the reciprocal influences became apparent. Hence, we extended the LAI parameter list and identified 27 LAI-AET parameters for evaluation of the significance of LAI on the AET estimation in SWAT-T.

We applied the elementary effects method to evaluate the parameter sensitivity to understand the parameter interactions in SWAT-T with observed LAI data. Previous efforts to assess the sensitivity of LAI parameters have focused on a relative sensitivity index (Khanal and Parajuli, 2014; Nantasaksiri et al., 2021). In the present study, the sensitivity of the comprehensive

set of 27 LAI-AET parameters is quantified with the elementary effects method in SWAT-T for the first time. Before, the Morris screening has been used for the sensitivity analysis of SWAT model parameters only with respect to discharge (Xiang et al., 2022; Abbas et al., 2024). With the application of the Morris screening, a ranking of the sensitivity of the parameters is determined in the present study. The most influential parameters on LAI are T_BASE, PHU, DLAI, and BLAI independent of the land cover type or the PET method. Moreover, SOL_RD is the parameter with the highest influence on the other parameters.

Its influence is significant because it defines the root depth within the soils which in turn determines the plant water uptake and thus the growing efficiency. The impact of SOL_RD is particularly meaningful in the forested region where the uptake of plants is high and the roots are growing deep. The sensitivities reported in Khanal and Parajuli (2014) are highest for the parameters DLAI, BIO_E, BLAI and SOL_RD. Nantasaksiri et al. (2021) identified the parameters BIO_E, HVSTI, BLAI, LAIMX$_2$, and DLAI to be the most sensitive. The findings in both studies are coherent with our results, albeit the missing investigation of





some of the most sensitive parameters, e.g., T_BASE and PHU. Moreover, we apply a global sensitivity measure while Khanal
and Parajuli (2014) and Nantasaksiri et al. (2021) have used a local measure (relative sensitivity index). Local measures are
however limited if the model response is nonlinear (Saltelli et al., 2008) which is the case for the LAI-AET parameters with
respect to observed LAI (Fig. 4). Thus, we were able to detect and address the non-linearity of the LAI-AET parameters with
the elementary effects method in the present study.

The field measurements used in this study are derived from hemispherical photographs and satellite-based corrections. Such
assorted LAI data can be subject to uncertainties (Fang et al., 2019). To address potential shortcomings of the LAI observations,
we additionally evaluated the LAI estimation regarding satellite-based GLASS-LAI. For both land cover types, the performance
of the LAI prediction is thereby accurate independent of the PET method. We applied the GLASS-LAI data since it is reliable
in different regions worldwide (Liang et al., 2014) and robust to noise and uncertainties satellite-based vegetation can be
susceptible to in tropical regions (Viovy et al., 1992; Atkinson et al., 2012). The dual consideration of both observed and
GLASS-LAI data assured the comprehensive LAI evaluation in the present study.

## 4.2   Optimization and benchmarking of the AET modelling with observed AET on the footprint scale

The model extent of the grassland region (2300 m$^2$) represents well the actual footprint size of 4000 m$^2$ estimated by Mamadou
et al. (2014). The footprint for the forested region is seasonally varying and can be up to 60000 m$^2$ (Mamadou et al., 2014).
Generally, the source area of AET in EC systems can fluctuate over the year (Kim et al., 2018) because of, for example, the
wind direction where the windrose can influence the extent of the footprint (Chen et al., 2009; Chu et al., 2021). Since the
model extents of SWAT-T are constant for the modelling period and the necessary data was not available, we approximated
the model scale to be representative for the footprint for the whole season according to Chu et al. (2021). The main objective
of the present study is the thorough evaluation of the vertical fluxes (AET) based on the LAI-AET interaction in SWAT-T. In
SWAT/SWAT-T, the vertical fluxes (AET) are computed on the HRU level. Hector et al. (2018) investigated the same regions
using a physically based model for the critical zone (ParFlow-CLM) and also concluded the significance of vegetation on the
AET estimation which is coherent to our findings.

In the present study, we also investigated if a detailed LAI modelling disregarding AET can predict reliable AET estimates
in SWAT-T. We showed that for both land cover types the LAI optimization also facilitates an adequate, behavioral modelling
of AET with acceptable KGE values. However, the evaluation of the model performance where only the values of an efficiency
metrics (e.g., KGE) are considered can be misleading because the explanatory power of the model is missing (Schaefli and
Gupta, 2007; Knoben et al., 2019). The information if a modelling approach is applicable or should be rejected and the
assessment of the strengths and deficiencies of the modelling approach is not covered in pure values of an efficiency metrics
(e.g., KGE) (Knoben et al., 2019). To address these shortcomings of an exclusive KGE value evaluation, we applied the
benchmark test proposed by Seibert et al. (2018). The comparison of modelling approaches, such as the single LAI optimization
with upper and lower benchmark levels, facilitated the assessment if a detailed LAI modelling (single LAI optimization) can
improve the LAI prediction in SWAT-T. Thereby, the benchmarking unfolded that the significance of a thorough LAI modelling
is more pronounced in the forested than in the grassland region.





## 4.3 Impacts of the model structure on the AET estimation

On the daily time-step, the temporal dynamics of simulated AET fit adequately to the observed AET pattern in the dry and wet season for all three PET methods. Thereby, the application of PET-PM outperforms PET-HG and PET-PT. Generally, the PET-PM application is more physically complex than PET-HG and PET-PT but also requires more input data. The computation of PET-HG and PET-PT relies on empirically delineated coefficients, e.g., $H_0$ and $\alpha_{pet}$, respectively. In PET-PM, terms for different properties of the land-atmosphere interaction are implemented, such as vapor pressure or the canopy $r_c$ and aerodynamic

resistance $r_a$. In PET-PT however, the aerodynamic term $\alpha_{pet}$ is modelled with a constant coefficient which is 1.28 (Neitsch et al., 2011). Moreover, the partitioning of PET into potential plant transpiration and soil evaporation is threshold based in PET-HG and PET-PT. While PET-PM estimates the potential transpiration using the PM equation where $r_c$ and $r_a$ are dependent on the modelled LAI modelling, the partitioning of PET implemented in PET-PT and PET-HG is based on the threshold $LAI > 3.0$. Hence, the significance of a detailed LAI modelling in these methods is less impactful on plant transpiration. For

the forested region, the LAI modelling (single LAI optimization disregarding AET) can still predict the AET adequately. The influence of LAI estimation is less substantial in the grassland region where the lower benchmark (random sampling) outperforms the single LAI optimization (observed and GLASS-LAI). The resulting median performance of the samples can yield an acceptable AET prediction where the KGE performance is better than compared to the LAI optimization. Thereby, the physical representation of AET and soil evaporation is however limited. Although the resulting parameters of the corresponding median

AET runs facilitate a satisfactory AET modelling, they do not account for the physical behavior of AET fluxes in the study site. For example, the ESCO parameter is determined to 0.91 in the case of grassland and PET-PM resulting in high soil evaporation rates (see Tab. A1). We also observed high interception rates (high CAN_MX values) which are modelled for PET-HG and PET-PT applications. The interception rate is translated to the eventual AET sum together with actual plant transpiration and soil evaporation. Since CAN_MX is less sensitive for LAI calibration, it has not been explicitly adjusted in the LAI-obs and

LAI-GLASS optimization. The AET estimation with SWAT-T can be susceptible to interception (López-Ramírez et al., 2021) because of its simplified model implementation (Neitsch et al., 2011). With a high values for the ESCO and CAN_MX parameters, considerable soil evaporation rates are computed in SWAT-T. However in the perennially vegetated grassland study site, the greater share of plant transpiration to soil evaporation for AET should be considered (Mamadou et al., 2016). Overall, the more simple approaches PET-HG and PET-PT can still yield adequate AET outputs (Archibald and Walter, 2014), although

PET-PM offers a more physically sound depiction of the LAI-AET interaction.

  In previous studies, similar accurate AET performances for the PET-PM application have been observed for a forested region (Ferreira et al., 2021) as well as for a grassland region (Qiao et al., 2022) regarding a comparison with AET from EC systems. An improvement of the AET estimation with SWAT-T using EC systems has also been demonstrated for PET-HG, PET-PT, and PET-PM on the HRU scale in López-Ramírez et al. (2021). The comparison of the annual budgets for AET fits thereby

best for PET-HG (López-Ramírez et al., 2021). However, no coupled LAI-AET parametrization has been considered. We were able to address the relevance of the coupled LAI-AET parametrization and thus demonstrated the best overall performance for PET-PM.



## 4.4 Outlook

The elementary effects were computed based on the whole period for which measured LAI data is available. SWAT-T divides
the plant growth in four phases (start of growing, maturity, leaf senescence, dormancy). A time-varying sensitivity analysis of
the LAI-AET parameters with respect to the plant growth phases has not been conducted but should be done in future work.
Possible applications should explore approaches such as dynamic identifiability analysis (Wagener et al., 2003) or wavelet
based methods (Chiogna et al., 2024). With these time-varying approaches, the understanding of the LAI-AET parameter
interaction can be further improved based on different scales and temporal periods.

Additionally, the plant growth modelling in SWAT-T also accounts for carbon dioxide levels in the atmosphere. Carbon
cycles play a pivotal role in the assessment of climate change impacts on watersheds, notwithstanding their impact on the plant
phenology (Gerten et al., 2004). Dynamic vegetation modelling is essential for the prediction of land-atmosphere interaction.
However, dynamic vegetation modelling is not always thoroughly addressed in climate change studies (Stephens et al., 2020).
On the African continent, temperature and carbon dioxide variations are predicted in the near and far future (Kusangaya et al.,
2014) and their influence on the hydrological processes are widely discussed (Akoko et al., 2021; Chawanda et al., 2024).
We highlighted the coupled LAI-AET relevance and ranked the most sensitive parameters, such as the T_BASE parameter in
SWAT-T. Changes in temperature are hence crucial for the LAI-AET interaction and can have an essential influence on plant
phenology and AET. Future work should pay attention to variations in the carbon dioxide levels and investigate their influence
on the LAI-AET interaction with SWAT-T.

We showed that the LAI-AET modelling of SWAT-T for approximated footprints is applicable for perennially vegetated
regions in Western Africa. However the actual, time-varying footprint scales should be considered in future work to further
improve the understanding of the LAI-AET interaction in SWAT-T. The studies should be extended cover multiple sites where
both AET and LAI data is available, e.g., the perennially vegetated stations of FLUXNET network (Friend et al., 2007).

    The present study mainly focuses on LAI as a vegetation attribute. In SWAT/SWAT-T, the canopy height is modelled, too.
The canopy height can have an impact on the PET estimation, e.g., in the application of PET-PM where the canopy resistance
$r_c$ is a function of the canopy height. Moreover, EC systems can also offer other relevant attributes of the vegetation-AET
interaction, such as derivations of the aerodynamic conductance, surface conductance, water vapor and heat fluxes, or the
evaporative fraction (Mamadou et al., 2016). These attributes not only improve the physical understanding of the vegetation-
AET interaction, but can also be valuable to inform hydrological modelling (Hector et al., 2018). We focused on the application
of LAI because (i) it is a key vegetation attribute in SWAT-T and (ii) global products of LAI are available making our approach
transferable to other regions. Since the seasonal dynamics of both, forest and grassland vegetation (LAI) is modelled accurately,
we postulate that the approaches of this study can be transferred to other plant and crop types and scales. With the consideration
of a coupled LAI-AET parametrization, the quantification of biomass or crop yield for other plant species can be addressed.
The application of satellite-based LAI data, e.g., GLASS-LAI, can support the plant growth and AET modelling on larger
scales, too, i.e. on the catchment scale.



## 5 Conclusion

The broad implication of this research is the presentation of a comprehensive LAI-AET parameter evaluation to model both, LAI and AET with an ecohydrological model. We highlighted the relevance of a coupled LAI-AET parameter estimation in SWAT-T. Although the impact of LAI parameters on the AET prediction can be low, substantial influences can be observed on the AET dynamics. The impact of the LAI parameters on AET is particularly high at the end of the wet season and the beginning of the dry season where the plant growth phase shifts from plant maturity to leaf senescence. Moreover, water stress on plant growth resulting from the AET estimation can be decisive and should be considered for a comprehensive LAI modelling. We conclude that the relevance of a coupled LAI-AET parameter estimation indicates that a stepwise modelling approach (e.g., first LAI, afterwards AET) requires a careful review of the simulated LAI after the AET parameters were estimated. The analysis of the elementary effects method demonstrates that the majority of LAI parameters behave non-linearly if compared to observed LAI data. The most sensitive parameters for LAI modelling are those associated as LAI parameters. Yet, the Morris screening also indicates a meaningful contribution of soil parameters. The ranking further illustrates an independence of the LAI parameters to the land cover type (forest and grassland).

The multi-objective optimization with SCE-UA algorithm results in accurate estimations of both, LAI and AET for all PET-methods and land cover types. SWAT-T has been proven to be applicable also on the footprint scale in Western Africa. Although the simpler PET-HG and PET-PT methods facilitate satisfactory modelling results, the application of PET-PM method outperforms these methods for the LAI and AET estimation in the forested and grassland region. Moreover, our work demonstrates that an adequate estimation of AET can be obtained if the LAI-AET parameters are only optimized with respect to LAI data (and disregarding AET data) for forest and grassland regions. The benchmark test illustrates an enhancement of the AET prediction for all three PET-methods (PET-HG, PET-PT, PET-PM) compared to the lower benchmark level. This is particularly noteworthy for data-scarce regions where no field measurements of AET are available. Even if no observed LAI data for a forested region is available, practitioners and researchers can optimize the LAI-AET parameters using remotely-sensed LAI data and still achieve reliable AET estimations. In the grassland region, the resulting AET prediction from the LAI optimization is adequate, too. However, the lower benchmark depicts a better performance for the grassland site. The good result of the lower benchmark is obtained from the median KGE performance of a large number of parameter samples (1000 runs). Single parameter changes, mean or the default model parameter values of the SWAT/SWAT-T crop data base do not necessarily facilitate a satisfactory AET prediction. Overall, the LAI-AET parameter optimization for grassland yields a sufficient AET performance. Nevertheless, its role in the AET estimation is less important than for forested regions.

Finally, we stress the importance of opting for a coupled parameter estimation to understand the LAI-AET interaction and to improve the land-atmosphere simulation in hydrological modelling. The performance comparison of modelled AET confirms that a detailed analysis of plant growth is essential. The highlighted relevance of the LAI-AET interaction is particularly meaningful for a thorough quantification of hydrological processes. The proposed consideration of a coupled LAI-AET modelling is not limited to footprint modelling in Western Africa. It is transferable to other regions and hence facilitates an advancement for the comprehensive assessment of the terrestrial water cycle on multiple scales for water resources management.





# Appendix A: Appendix

## A1 Appendix A1 - Figures

**Figure A1.** Distribution of variations in AET or LAI for the one-at-a-time parameter changes for PET-HG. The distributions are clustered in uniform intervals (size = 0.625) of the observed time series for AET [mm] or LAI [m²/m²]. The x-axis indicates the observed AET [mm] and LAI [m²/m²] values. The y-axis represents the difference between observed and simulated values with $\Delta Y = Obs - Sim$ regarding AET [mm] or LAI [m²/m²]. A perfect fit is indicated with the dashed line for $\Delta Y = 0$. Positive and negative values show an underestimation and overestimation of the simulated values, respectively. The distributions (violin plots) are created based on Karvelis (2024).



**Figure A2.** Distribution of variations in AET or LAI for the one-at-a-time parameter changes for PET-PT. The distributions are clustered in uniform intervals (size = 0.625) of the observed time series for AET [mm] or LAI [m$^2$/m$^2$]. The x-axis indicates the observed AET [mm] and LAI [m$^2$/m$^2$] values. The y-axis represents the difference between observed and simulated values with $\Delta Y = Obs - Sim$ regarding AET [mm] or LAI [m$^2$/m$^2$]. A perfect fit is indicated with the dashed line for $\Delta Y = 0$. Positive and negative values show an underestimation and overestimation of the simulated values, respectively. The distributions (violin plots) are created based on Karvelis (2024).

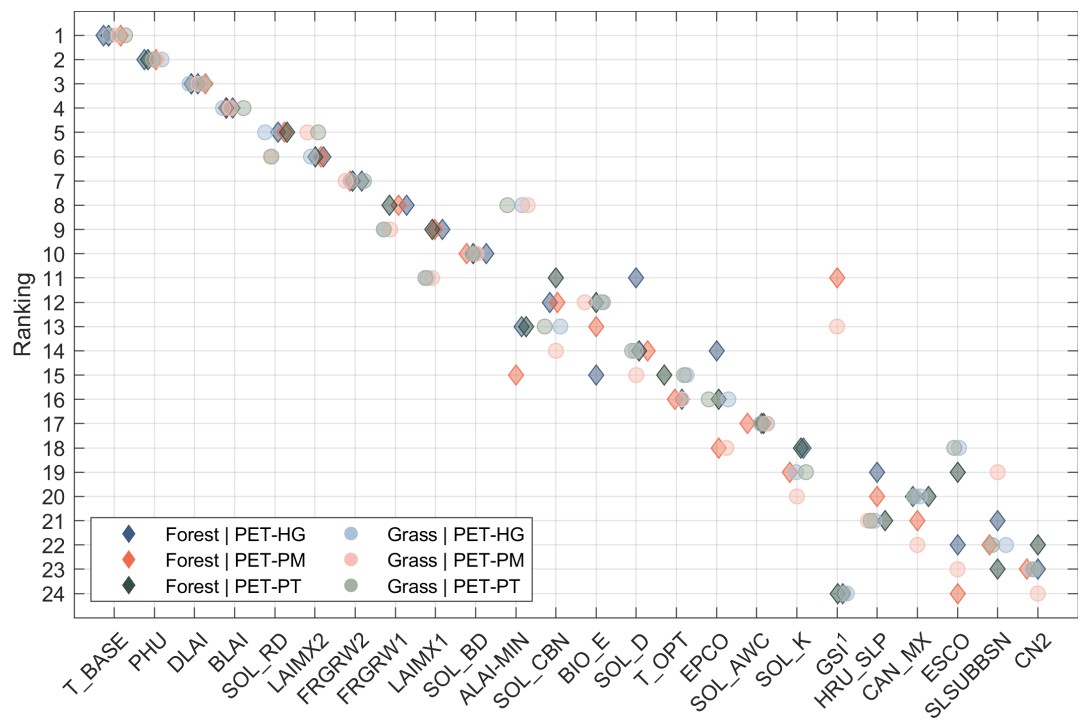

**Figure A3.** Ranking of the LAI-AET parameter sensitivity for the three PET methods and two land cover types with respect to observed LAI. [1]Parameter GSI is only accounted for when PET-PM is used.

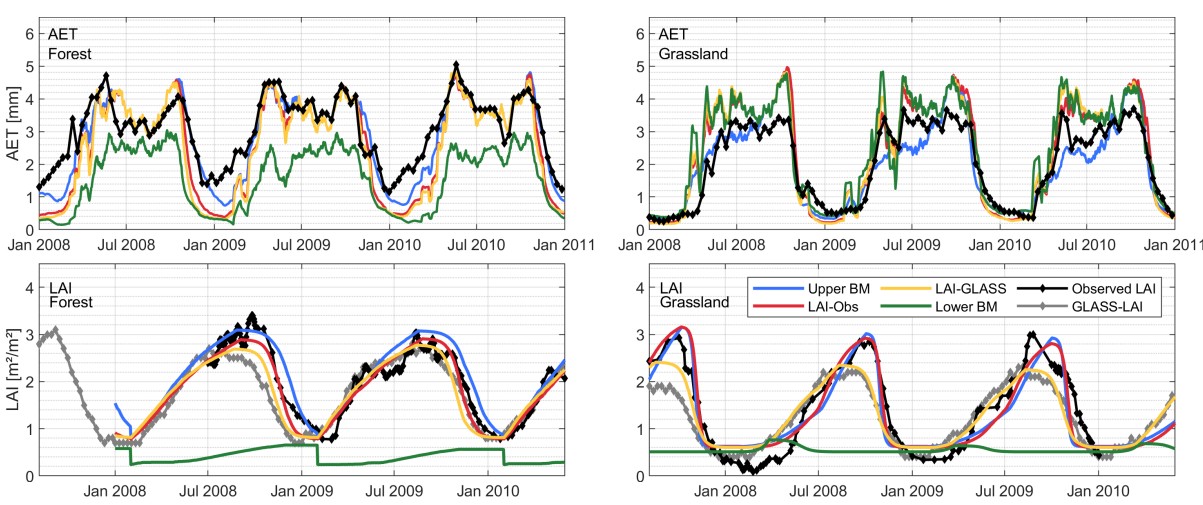

**Figure A4.** Time series of simulated and observed AET for the forested and grassland region. The simulated AET and LAI for the four benchmark elements are displayed for PET-PT.





## A2 Appendix A2 - Supplementary equations

Water stress $wstrs$ is calculated as:

$$wstrs = 1 - \frac{E_{t,act}}{E_t} = 1 - \frac{w_{actualup}}{E_t}, \tag{A1}$$

where $E_{t,act}$ is the actual transpiration, $E_t$ is the potential plant transpiration, and $w_{actualup}$ is the total water uptake. $w_{actualup}$ is computed based on the amount of water in the soil layer and the water content at the wilting point (for details refer to Neitsch et al. (2011)).

Temperature stress $tstrs$ is calculated as:

$$tstrs = \begin{cases} 1, & \text{if } T_{av} \leq T_{base} \\ 1 - exp\left(\frac{-0.1054 \cdot (T_{opt} - T_{av})^2}{(T_{av} - T_{base})^2}\right), & \text{if } T_{base} < T_{av} \leq T_{opt} \\ 1 - exp\left(\frac{-0.1054 \cdot (T_{opt} - T_{av})^2}{(2 \cdot T_{opt} - T_{av} - T_{base})^2}\right), & \text{if } T_{opt} < T_{av} \leq 2 \cdot T_{opt} - T_{base} \\ 1, & \text{if } T_{av} > 2 \cdot T_{opt} - T_{base} \end{cases} \tag{A2}$$

where $T_{av}$ is the mean air temperature for the day, $T_{base}$ is the base temperature of the plant for growth, and $T_{opt}$ is the optimal temperature of the plant for growth.

Nitrogen stress $nstrs$ is calculated as:

$$nstrs = 1 - \frac{\phi_n}{\phi_n + exp\left(3.535 - 0.02597 \cdot \phi_n\right)}, \tag{A3}$$

where $\phi_n$ is a scaling factor for nitrogen stress computed with the actual and optimal mass of nitrogen stored in the plant
material $bio_N$ and $bio_{N,opt}$, respectively:

$$\phi_n = 200 \cdot \left(\frac{bio_N}{bio_{N,opt}} - 0.5\right) \tag{A4}$$

Phosphor stress $pstrs$ is calculated as:

$$pstrs = 1 - \frac{\phi_p}{\phi_n + exp\left(3.535 - 0.02597 \cdot \phi_p\right)}, \tag{A5}$$

where $\phi_p$ is a scaling factor for phosphorous stress computed with the actual and optimal mass of phosphorus stored in the
plant material $bio_P$ and $bio_{P,opt}$, respectively:

$$\phi_p = 200 \cdot \left(\frac{bio_P}{bio_{P,opt}} - 0.5\right). \tag{A6}$$

The Kling-Gupta efficiency $KGE$ is calculated as:

$$KGE = 1 - \sqrt{(r-1)^2 + \left(\frac{\sigma_{sim}}{\sigma_{obs}} - 1\right)^2 + \left(\frac{\mu_{sim}}{\mu_{obs}} - 1\right)^2}, \tag{A7}$$

where $r$ is the linear correlation between observations and simulations, $\sigma_{sim}$ and $\sigma_{obs}$ are the standard deviation of the simu-
lations and observations, respectively, and $\mu_{sim}$ and $\mu_{obs}$ are the mean value for the simulations and observation, respectively.





## A3 Appendix A3 - Final parameters

**Table A1.** List of final parameters for the multi-objective (LAI & AET), observed LAI, and GLASS-LAI optimization for the forested region. The units of the parameters are excluded for readability. They are given in Tab. 2.

| Parameter | LAI & AET | | | Observed LAI | | | GLASS-LAI | | | Lower Benchmark | | |
|---|---|---|---|---|---|---|---|---|---|---|---|---|
| | HG | PM | PT | HG | PM | PT | HG | PM | PT | HG | PM | PT |
| BIO_E | 24.13 | 21.31 | 21.45 | 16.77 | 20.14 | 19.11 | 18.78 | 17.42 | 21.64 | 21.55 | 34.29 | 20.35 |
| BLAI | 4.44 | 4.35 | 4.57 | 3.78 | 4.42 | 4.54 | 4.80 | 3.89 | 4.32 | 9.09 | 7.68 | 3.93 |
| $FRGRW_1$ | 0.18 | 0.15 | 0.16 | 0.19 | 0.15 | 0.16 | 0.12 | 0.16 | 0.15 | 0.06 | 0.20 | 0.18 |
| $LAIMX_1$ | 0.18 | 0.21 | 0.17 | 0.21 | 0.17 | 0.17 | 0.22 | 0.23 | 0.21 | 0.16 | 0.21 | 0.20 |
| $FRGRW_2$ | 0.81 | 0.72 | 0.72 | 0.64 | 0.68 | 0.75 | 0.75 | 0.76 | 0.73 | 0.60 | 0.82 | 0.36 |
| $LAIMX_2$ | 0.74 | 0.68 | 0.72 | 0.73 | 0.74 | 0.66 | 0.61 | 0.74 | 0.65 | 0.69 | 0.71 | 0.31 |
| DLAI | 0.58 | 0.63 | 0.53 | 0.57 | 0.56 | 0.61 | 0.58 | 0.58 | 0.60 | 0.25 | 0.44 | 0.33 |
| T_OPT | 40.25 | 39.93 | 39.77 | 40.12 | 38.92 | 39.19 | 38.55 | 42.24 | 41.26 | 45.16 | 35.75 | 23.00 |
| T_BASE | 14.03 | 14.46 | 15.77 | 14.15 | 15.22 | 14.75 | 15.28 | 14.70 | 14.53 | 11.78 | 8.80 | 19.08 |
| ALAI_MIN | 0.88 | 0.77 | 0.79 | 0.82 | 0.82 | 0.78 | 0.70 | 0.80 | 0.81 | 0.36 | 0.25 | 0.24 |
| PHU | 4336 | 4075 | 4087 | 4337 | 4082 | 4013 | 3776 | 3877 | 3956 | 5607 | 4783 | 6110 |
| GSI | 0.006 | 0.005 | 0.006 | 0.007 | 0.005 | 0.007 | 0.005 | 0.005 | 0.005 | 0.001 | 0.008 | 0.005 |
| CAN_MX | 9.0 | 59.3 | 37.54 | 60.1 | 61.0 | 48.8 | 29.69 | 61.66 | 54.60 | 76.90 | 92.74 | 7.31 |
| ESCO | 0.64 | 0.50 | 0.51 | 0.55 | 0.54 | 0.47 | 0.43 | 0.53 | 0.39 | 0.36 | 0.38 | 0.86 |
| EPCO | 0.88 | 0.47 | 0.84 | 0.59 | 0.33 | 0.53 | 0.37 | 0.43 | 0.41 | 0.63 | 0.03 | 0.19 |
| HRU_SLP | 0.005 | 0.006 | 0.005 | 0.005 | 0.006 | 0.005 | 0.003 | 0.003 | 0.004 | 0.06 | 0.10 | 0.53 |
| SLSUBBSN | 35.19 | 28.9 | 30.06 | 33.75 | 30.11 | 30.13 | 25.12 | 39.14 | 26.54 | 30.98 | 69.90 | 25.75 |
| CN2 | 39.58 | 46.91 | 44.92 | 47.7 | 44.65 | 45.63 | 52.80 | 41.02 | 42.72 | 39.69 | 43.72 | 58.20 |
| SOL_AWC[a] | 1.76 | 1.48 | 1.6 | 1.31 | 0.64 | 0.83 | 1.25 | 0.74 | 0.62 | 0.73 | 0.63 | 1.12 |
| SOL_BD[a] | 0.59 | 0.69 | 0.82 | 1.13 | 0.64 | 0.83 | 1.02 | 0.46 | 0.50 | 0.02 | 0.44 | -0.47 |
| SOL_CBN[a] | 0.11 | 0.41 | 0.49 | 0.73 | 0.60 | 0.33 | 0.27 | 0.63 | 1.10 | -0.26 | -0.09 | 1.00 |
| SOL_K[a] | 0.97 | 0.57 | 0.68 | 0.31 | 0.65 | 0.51 | 0.70 | 0.68 | 1.10 | -0.11 | 1.06 | 1.79 |
| SOL_RD | 1583 | 1356 | 1438 | 1160 | 1446 | 1382 | 768 | 1219 | 1475 | 1529 | 852 | 1688 |
| SOL_D[b] | 3656 | 3359 | 3237 | 3263 | 3158 | 3070 | 3087 | 3518 | 2838 | 1631 | 1716 | 3380 |
| GW_REVAP | 0.12 | 0.11 | 0.12 | 0.06 | 0.10 | 0.10 | 0.11 | 0.08 | 0.13 | 0.15 | 0.93 | 307 |
| RCHRG_DP | 0.44 | 0.48 | 0.49 | 0.33 | 0.30 | 0.47 | 0.60 | 0.57 | 0.59 | 0.18 | 0.98 | 0.14 |
| REVAPMN | 743 | 999 | 1025 | 885 | 1155 | 961 | 646 | 798 | 1062 | 307 | 1093 | 313 |

[a]Relative parameter changes: $para_{new} = para_{original} + para_{original} \cdot para_{change}$ ; [b]Lowest soil layer depth





**Table A2.** List of final parameters for the multi-objective (LAI & AET), observed LAI, and GLASS-LAI optimization for the grassland region. The units of the parameters are excluded for readability. They are given in Tab. 2

| Parameter | LAI & AET | | | Observed LAI | | | GLASS-LAI | | | Lower Benchmark | | |
|---|---|---|---|---|---|---|---|---|---|---|---|---|
| | HG | PM | PT | HG | PM | PT | HG | PM | PT | HG | PM | PT |
| BIO_E | 29.53 | 23.72 | 27.49 | 20.09 | 20.12 | 20.88 | 21.12 | 20.61 | 21.92 | 31.23 | 35.38 | 23.71 |
| BLAI | 5.01 | 4.76 | 4.77 | 4.14 | 4.34 | 4.10 | 3.95 | 4.13 | 4.46 | 3.17 | 4.00 | 2.48 |
| FRGRW$_1$ | 0.30 | 0.28 | 0.27 | 0.29 | 0.30 | 0.30 | 0.23 | 0.25 | 0.26 | 0.09 | 0.26 | 0.25 |
| LAIMX$_1$ | 0.07 | 0.06 | 0.06 | 0.05 | 0.05 | 0.05 | 0.11 | 0.12 | 0.13 | 0.28 | 0.12 | 0.07 |
| FRGRW$_2$ | 0.75 | 0.63 | 0.74 | 0.63 | 0.62 | 0.63 | 0.56 | 0.53 | 0.57 | 0.58 | 0.32 | 0.50 |
| LAIMX$_2$ | 0.74 | 0.74 | 0.77 | 0.76 | 0.77 | 0.77 | 0.87 | 0.85 | 0.88 | 0.50 | 0.59 | 0.64 |
| DLAI | 0.82 | 0.81 | 0.83 | 0.78 | 0.78 | 0.80 | 0.61 | 0.61 | 0.61 | 0.21 | 0.82 | 0.31 |
| T_OPT | 39.83 | 40.91 | 40.52 | 39.47 | 40.05 | 40.28 | 39.57 | 39.88 | 39.10 | 49.25 | 49.14 | 46.53 |
| T_BASE | 12.80 | 12.92 | 12.25 | 12.82 | 12.96 | 12.66 | 13.85 | 13.81 | 13.92 | 0.34 | 8.91 | 7.0 |
| ALAI_MIN | 0.62 | 0.62 | 0.62 | 0.60 | 0.60 | 0.61 | 0.60 | 0.60 | 0.60 | 0.85 | 0.21 | 0.51 |
| PHU | 4073 | 3993 | 4132 | 4120 | 4057 | 4133 | 4027 | 4074 | 4040 | 2507 | 5902 | 3738 |
| GSI | 0.006 | 0.003 | 0.006 | 0.005 | 0.006 | 0.006 | 0.005 | 0.005 | 0.006 | 0.006 | 0.005 | 0.002 |
| CAN_MX | 0.15 | 16.50 | 0.05 | 40.01 | 39.02 | 36.46 | 43.36 | 39.10 | 36.00 | 19.72 | 74.48 | 88.68 |
| ESCO | 0.69 | 0.63 | 0.58 | 0.53 | 0.51 | 0.48 | 0.48 | 0.61 | 0.48 | 0.59 | 0.86 | 0.91 |
| EPCO | 0.44 | 0.23 | 0.50 | 0.30 | 0.39 | 0.34 | 0.22 | 0.22 | 0.27 | 0.88 | 0.36 | 0.52 |
| HRU_SLP | 0.007 | 0.005 | 0.005 | 0.005 | 0.004 | 0.005 | 0.005 | 0.005 | 0.006 | 0.18 | 0.67 | 0.99 |
| SLSUBBSN | 26.80 | 29.07 | 28.85 | 30.60 | 27.70 | 26.30 | 30.13 | 30.19 | 25.17 | 81.62 | 63.86 | 35.67 |
| CN2 | 41.86 | 41.91 | 44.52 | 43.43 | 44.87 | 42.90 | 46.51 | 44.70 | 45.93 | 56.60 | 36.80 | 87.56 |
| SOL_AWC[a] | 0.27 | 0.22 | 0.98 | 0.91 | 0.67 | 0.58 | 0.34 | 0.57 | 0.44 | 0.85 | 0.61 | 1.28 |
| SOL_BD[a] | 1.14 | 0.98 | 0.90 | 1.03 | 0.69 | 0.77 | 0.92 | 0.73 | 0.80 | 0.43 | -0.44 | 1.72 |
| SOL_CBN[a] | 1.18 | 1.11 | 1.09 | 0.86 | 0.95 | 1.01 | 1.48 | 1.28 | 1.36 | 1.70 | -0.21 | -0.42 |
| SOL_K[a] | 1.26 | 0.91 | 0.71 | 0.86 | 0.91 | 0.65 | 0.85 | 0.80 | 0.32 | 0.04 | 0.07 | 0.10 |
| SOL_RD | 1166 | 1189 | 1098 | 1294 | 1339 | 1341 | 1399 | 1284 | 1103 | 1521 | 905 | 720 |
| SOL_D[b] | 2800 | 2771 | 3159 | 2957 | 3013 | 3080 | 2949 | 3110 | 2876 | 2326 | 4892 | 1658 |
| GW_REVAP | 0.10 | 0.12 | 0.12 | 0.11 | 0.12 | 0.11 | 0.11 | 0.11 | 0.12 | 0.12 | 0.06 | 0.16 |
| RCHRG_DP | 0.27 | 0.56 | 0.50 | 0.46 | 0.47 | 0.55 | 0.46 | 0.45 | 0.53 | 0.73 | 0.48 | 0.08 |
| REVAPMN | 1034 | 977 | 799 | 1084 | 1101 | 1041 | 910 | 1067 | 1237 | 303 | 467 | 343 |

[a]Relative parameter changes: $para_{new} = para_{original} + para_{original} \cdot para_{change}$ ; [b]Lowest soil layer depth



*Author contributions.* FM, TS, FA, YT, JMC, MD reviewed and edited the manuscript; FM, TS, YT, FA, MD conceived and designed the study; FM, FA, JMC acquired the data; FM, TS performed the data analysis and model development and simulations; FM, TS, FA, YT, JMC, MD evaluated the simulations and models. FM wrote the manuscript draft.

*Competing interests.* The authors declare that they have no conflict of interest.

*Acknowledgements.* The authors want to thank the BMBF ("Bundesministerium für Bildung und Forschung") for the funding of the FURI-FLOOD research project ("Current and future risks of urban and rural flooding in West Africa", Grant No.: 01LG2086B). We would like to thank our partners involved in the FURIFLOOD project for their support. This work was further supported by the European Union's Horizon Europe research and innovation program as part of the UAWOS project ("Unmanned Airborne Water Observing System", Grant Agreement
No.: 101081783). We would also like to thank the team of the AMMA-CATCH network for their support and their quick responses in case of any queries.



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
