# Peer review of "The Significance of the Leaf-Area-Index on the Evapotranspiration Estimation in SWAT-T for Characteristic Land Cover Types of Western Africa"

_Hydrology and Earth System Sciences, 2024_

## Author Response (AR1)

**Author's response of Merk et al. (Hess-2024-131)**

A point-by-point response to the reviews including a list of all relevant changes made in the manuscript.

**Point-by-point replay on RC1**

(https://doi.org/10.5194/hess-2024-131-RC1)

Dear Reviewer RC1,

On behalf of all authors, I want to again thank you for your comprehensive feedback on our work. Your remarks are valuable for the publication and will improve its relevance for the community. The following point-by-point replay follows our first response which can be found here https://doi.org/10.5194/hess-2024-131-AC1.
A list of relevant changes in the latest manuscript is added at the end of this replay.

With kind regards
Fabian Merk

Please note that the original review comments are colored in black and our answers are highlighted in blue.

**Major comments:**

- The authors described that forest and grassland were defined with FRSD and RNGE land use codes from the SWAT crop data based on lines 203-204. However, it is not clear if their parametrization (i.e., initial values and calibration range) were adapted for tropical vegetation characteristics. In particular, the final values for some parameters like T_OPT (optimal temperature for plant growth) may not be realistic for the study region (see Alemayehu et al., 2017 results). For example, Tables A1 and A2 show that T_OPT final values for grassland and forest vegetation were larger than 35 Celsius degrees for all approaches and PET methods, except for forest in lower benchmark analysis. Thus, are realistic the final parameter values (e.g., T_OPT, T_BASE, RCHRG_DP)?

Answer:
Thank you for questioning the final parameters. Following your remarks and those in https://doi.org/10.5194/hess-2024-131-RC2, we simulated all 24 benchmark setups again. Indeed, the ranges for T_OPT are too high. We constrained the ranges for T_OPT (20°C – 32°C) to keep the physical representation of the plant growth in the tropics for the second set of simulations. T_BASE is the most sensitive LAI-AET parameter because it governs the temperature stress equations on the actual plant growth in SWAT-T. In both versions, the optimal T_BASE is around 14°C which can be assumed for sub-humid zones.
In addition, we set the parameters for discharge (HRU_SLP, SLSUBBN, CN2) and groundwater (GW_REVAP, RCHRG_DP, REVAPMN) to constant values in the benchmarking. We used literature values from comparable studies. With these constant values, the focus on the LAI-AET parameters is guaranteed and the equifinality problem in hydrological modelling is not solved but addressed.

The new simulations show similar results to the first version and are presented in the new manuscript:
- Explanation for the constant parameters line 286-290
- Results in 3.3 Optimization and benchmark test of the LAI-AET modelling line 365

- The study highlights that "*if an optimization process only accounts for LAI but disregards AET data, its prediction of AET still yields an adequate performance with SWAT-T for all PET methods and land cover types*". This represents an important contribution due to the lack of AET ground observations in tropical regions. However:
  - Do you expect similar results for streamflow? Despite performance metrics showing that AET results are acceptable considering only LAI data, it is interesting to discuss what would be the performance of other water balance processes.

Answer:
AET is up to 70% of the water budget in this region (Mamadou et al., 2016). The rate of AET could be even higher for gallery forest. AET is hence a dominant hydrological process, particularly compared to discharge. In our study, we put a special focus on the dominant, vertical fluxes, i.e., on AET.
For the computation of streamflow, vertical and horizontal fluxes are substantial. At this point and without the specific data and analysis, we can only theoretically address the implications for streamflow. The prediction of AET facilitates an adequate spatial representativeness of the vertical fluxes. If the AET is for example predicted accurately for this type of AET dominant regions, most of the water, say 70 %, entering the system from precipitation is already accounted for the correct way. The remaining 30 % are split into storages, groundwater recharge, and discharge. For hydrological modelling in general and SWAT/SWAT-T in particular, other parameters than the LAI-AET parameters are decisive for the discharge computation, such as the CN number. We have proven that the CN parameter has less significance on LAI-AET. Still, it is significant for the discharge computation. It's thus not guaranteed that the discharge prediction will certainly improve with an LAI data only. Still, the spatial representation of the water balance can be improved because LAI and AET data is available on a spatial resolution. A model calibration where all model parameters are optimized according to one point (discharge at model outlet) often results in good metrics for discharge. Yet for these approaches, it is not guaranteed that the spatial representation of the hydrological processes make sense. More studies on the catchment scale are necessary if an evidence-based conclusion on the implications of LAI-AET modelling on the streamflow prediction is relevant.

We adjusted the paper accordingly to address the potential implications of the LAI-AET modelling on streamflow. The addition is narrowed down to try to keep the paper concise.
Changes are in line 482 to 488

- How would be the LAI-AET interactions using only remote-sensing datasets (e.g., AET and LAI from MODIS)?

Answer:
In the data acquisition and preprocessing phase, we figured that remotely-sensed products for LAI are generally poor in performance to mimic the vegetation growth in this region. Exceptions are the GLASS and the latest Copernicus products. The GLASS product is based on MODIS data but further filters the data using machine learning approaches. The comparison of GLASS and the observed LAI shows an adequate fit for our sites. Other products or previous versions of Copernicus were bad because of the cloud coverage and the low number of useful images. The GLASS and the latest Copernicus data are both acceptable and we can expect reasonable results also for other parts of this region with these products. The MODIS data often needs filtering and smoothing. Moreover, the performance of remotely-sensed AET data (like MODIS) varies globally. The spatial scale of MODIS AET data, e.g., the MOD16A2 version, is 500 m. For the spatial dynamics of AET, this scale might be too coarse and hence necessitates a careful application, especially with regard to the underlying land

cover and soil type. If both, the LAI and AET data from MODIS is acceptable for specific, vegetation-AET dominated sites, then similar findings to ours can be expected.

- The authors state that their results are transferable to other regions in Lines 543-544. However, it is necessary to provide further details regarding the generalization of their findings considering:
    o Heterogeneous land cover and soil properties. Indeed, the SWA-T model was set up with a single HRU in this study.
    o Do you expect similar results for larger watersheds (e.g., $10 – 100\ km^2$)?
    o Are the results similar for forested areas with a larger LAI (e.g., Amazon and Congo forests).
    o Although ET is mainly controlled by water limitations in tropical ecosystems, some mountain areas exhibit energy limitations. What are the implications of the study's finding for energy vs water limited ecosystems?

Answer:
Thank you for questioning the transferability of our results to other regions. Indeed, we can only anticipate a transferability based on the analysis we did in this study.
For larger watersheds (e.g., $10 – 100\ km^2$), the land cover and soil properties are decisive. As shown in our study, some soil parameters, particularly the rooting depth (SOL_RD) and bulk density (SOL_BD) (see Figure 5), are sensitive and hence can be important when modelling LAI. The soils in our study are loamy sands in the upper and sandy clay loam in the lower horizons (Mamadou et al., 2016). Hence for regions with similar soils, e.g., in the Sudanian zone, we anticipate comparable results.
For forested areas with larger LAI (e.g., Amazon and Congo forests), we can't predict results without specific data. However in the context of all the Sudanian areas the climate/vegetation/geology context is quite similar and our results should be representative for the whole region. The application of the predecessor of SWAT-T (see Strauch & Volk, 2012) has been applied in the Amazon region to model both LAI and AET. Given the LAI-AET interaction and its reciprocal dependency, i.e., LAI is a term of the Penman-Monteith equation, a sound modelling of LAI should improve the AET estimation also in areas of larger LAI values.
The implications of the LAI-AET interaction in energy vs. water limited ecosystems can only be theoretically answered based on our study. The footprint sites in our study are located in a water limited ecosystem where the annual PET is larger than the annual precipitation (see appendix in the manuscript). Theoretically, the Penman-Monteith PET formula considers energy fluxes and plant phenology of which LAI is an attribute. Hence, LAI can be expected to also have a meaningful role in energy limited ecosystems. Still, we lack explicit data and analysis based on this study to reason a transferability to energy limited ecosystems.
Overall, we can only anticipate the transferability of our results to other regions. Without discussed evidence, we can only assume the role of LAI on other ecosystems than those which are similar, e.g., in the Sudanian region. Still, the transferability is a very interesting research question, particularly the transition from the footprint to the catchment scale. We are currently working on a publication where we transfer the LAI-AET modelling and our findings from the footprints to the whole Bétérou catchment.
Ultimately, we adjusted the paper accordingly. We argue that this research question remains important for the hydrologic community, especially in AET dominated regions.

Changes: we moved the anticipated transferability from the 5. Conclusion to the 4.4. Outlook chapter.

**Minor comments:**

- Lines 33-40: Many SWAT studies also have used ET data from remote-sensing datasets such as MODIS (e.g., Qiao et al., 2022; Rajib et al., 2018)

Answer:
We added these sources in the new text version, lines 54-55 and line 471

- Lines 50-55: Additional SWAT-T applications: Hoyos et al., 2019

Answer:
We added these sources in the new text version; line 52

- Lines 85-90: What tool or software was used for sensitivity analysis? SWAT-CUP?

Answer:
We applied the elementary effects method, or Morris method, which is an approach to conduct a global sensitivity analysis. We followed the publication of Morris and scripted the functions in MATLAB on our own. The code needs to be cleaned, i.e., comments need to be added. We plan to publish it on a depository. It will be made available at least on demand. We will add a sentence to precise this methodology point.
Lines 249-250.

- Lines 180-185: Are savanna dynamics affected by cropland management?

Answer:
In the Nalohou region, the land cover is largely covered by characteristic mixture of crops and savannah grassland and fallows. As far as we observed in the field, the purely herbaceous plots are young fallow. When parcels are not cultivated, these change rapidly after some years more to bushy savannah. At the seasonal scale, this bushy savannah vegetation is not managed except for some limited wood harvesting. The Nalohou data are typical of croplands including young fallow with growing on burnt land (burning of fallows at the beginning of the dry season). We still assigned the crop type RNGE from SWAT/SWAT-T because it is a generic land use type for these kind of land covers (see Alemayehu et al., 2017).

- Figure 2b: What is the annual cycle of solar radiation, relative humidity, and wind speed?

Answer:
We added a figure to the appendix in the new text version, line 540

Figure 2c: Why does AET decrease during the wet season? Could this pattern suggest an energy-limited environment? Include observed LAI.
Answer:
Looking at the Mamadou's JGR paper (Fig. 3), net radiation is decreasing during the wet season, with big drops down to zero which automatically reduces other fluxes like sensible and latent heat fluxes. Additionally, the atmospheric demand is reduced because of high air humidity which was observed for the vapor pressure deficit in Mamadou et al. (2016). Moreover, AET is the sum of evaporation and transpiration. It is not expected AET follows strictly the LAI. We adjusted the manuscript accordingly.
Lines: 177-181

Line 222: Was the analysis also conducted for the grassland site?
Answer:
The objective of this evaluation is to show the relevance of the LAI-AET interaction. We also did it for the grassland site but observed highly similar findings to the forested region. We hence excluded the analysis from the paper to keep it concise. We argue that the relevance is clearly shown in the

forested region. Basically, the figures for the grassland are similar and the conclusion is identical to the forested region. Hence, we skipped this figure for keep the paper short and the focus on the LAI-AET sensitivities and benchmarking.

- Figure 3: Include panel letters (a, b, c) and supplementary figures for other parameters.

Answer:
We added panel letters to all figures in the new version, see all multi-panel figures in the manuscript.

- Line 315: What does mean "can be decisive"? Significant?

Answer:
We adjusted the manuscript accordingly, line 311.

- Figure 4: Please consider color-blind palettes (apply in all figures).

Answer:
We adjusted the manuscript (and its figures) accordingly. We used the suggestions from: https://www.nceas.ucsb.edu/sites/default/files/2022-06/Colorblind%20Safe%20Color%20Schemes.pdf

- Figures 6 and 7: Daily time-scale results. Include panel letters. Plotting percentage anomalies can provide a clear idea of AET and LAI differences between observations and simulations. Please also include an annual cycle plot to show the results for dry and wet seasons (Lines 445-447).

Answer:
Both suggestions are a very good idea to communicate modelling results. We created both, figures for percentage anomalies and annual cycles. Yet, we figured that the percentage anomalies figure is complex since it consists of 48 modelling setups (24 for AET, 24 for LAI). The annual cycle gives a clear overview of the overall fit of simulated to observed values. We hence included only the annual cycles to the new manuscript version. We suppose that the fit of simulated to observed data is meaningfully presented in the figures of the annual cycles.

- Lines 450-455: LAI is almost always lower than 3 for both forest and grassland vegetation (Figure 7).

Answer:
Yes, it is lower than 3. We know the footprint sites from field visits, from measurements (Ago et al., 2014), and from literature (e.g., Mamadou et al., 2016)

**List of most relevant changes**

Please note that we only list the most relevant changes. For changes concerning the minor comments, please refer to the answers we gave to the comments.

| Review comment | Change |
|---|---|
| Parameter choice and ranges | We simulated all 24 benchmark setups again with a new set of parameters. We defined constant parameter values for discharge, groundwater and geospatial parameters. The results of the new runs are similar. The respective changes are made in the manuscript. (line 286-290, Chapter Results) |
| Transferability to other climates and regions | Indeed, we are missing data and evidence to conclude the transferability of the LAI-AET approach to other regions.
But we can still anticipate the applicability and importance of LAI-AET modelling on the catchment scale, for an improved water balance estimation, and for other regions. Hence, we moved the transferability part from Conclusion to Outlook. |
| Figures | We adjusted all figures. We added panel letters (where necessary) and used color-blind palettes. In addition, we added relevant figures to the appendix such that the reader gets more insights |

References:

Ago, E. E., Agbossou, E. K., Cohard, J.-M., Galle, S., and Aubinet, M.: Response of $CO_2$ fluxes and productivity to water availability in two contrasting ecosystems in northern Benin (West Africa), Annals of Forest Science, 73, 483–500, https://doi.org/10.1007/s13595-016-0542-9, 2016

Mamadou, O., Galle, S., Cohard, J.-M., Peugeot, C., Kounouhewa, B., Biron, R., Hector, B., and Zannou, A. B.: Dynamics of water vapor and energy exchanges above two contrasting Sudanian climate ecosystems in Northern Benin (West Africa), Journal of Geophysical Research: Atmospheres, 121, https://doi.org/10.1002/2016JD024749, 2016

**Point-by-point replay on RC2**

(https://doi.org/10.5194/hess-2024-131-RC2)

Dear Reviewer RC2,

On behalf of all authors, I want to again thank you for your valuable feedback on our work. We are convinced that your remarks enhance the quality of our manuscript. Particularly the questioning of the final parameters was very valuable for us which we gratefully adjusted in the new version.
The following point-by-point replay follows our first response which can be found here https://doi.org/10.5194/hess-2024-131-AC2.
A list of relevant changes in the latest manuscript is added at the end of this replay.

With kind regards
Fabian Merk

Please note that the original review comments are colored in black and our answers are highlighted in blue.

**Major comments**

- The authors used 27 parameters in their optimizations, but only a few of these parameters significantly influence LAI simulations. Given the parameter compensation effect, also known as the equifinality problem, the influence and significance of LAI on ET could be overlooked. This is evident from the model performances reported in Table 5 for LAI. I suggest setting model parameters relevant to surface runoff and groundwater based on previously published values for the region or adjusting them using stream gauge data for the Bétérou catchment across all model setups. This approach could help to shed light on the influence of LAI on potential plant transpiration and, ultimately, on ET.
- The authors seem to rely heavily on the automatic optimization output without verifying the realism of the selected parameters space to at least reflect the difference between forest and grassland, which is problematic. For example, the calibrated CAN_MX (potential maximum canopy storage) ranges from 9 to 62mm, and the optimal temperature (T_OPT) values are well above 39°C. Additionally, the CN2 values for forest and grassland land cover types are closely comparable. This suggests an ill-defined model parameter range setup at the initial stage, resulting in poorly quantified model parameters. This issue needs to be addressed or justified.

Answer to both points:
Thank you for questioning the choice of parameters and their final values after optimization. It's true that we haven't addressed the equifinality problem in the first version. The focus of our study was a holistic analysis of relevant LAI-AET parameters. Hence, we considered all 27 parameters. Still, insensitive or transferable parameters can be held constant in the optimization process (or be excluded), to reduce the computational cost and to address the equifinality problem in hydrological modelling.
We followed your suggestion and adjusted the choice of parameters and the parameter ranges for selected parameters for the benchmarking. We defined physical meaningful ranges for T_OPT (20°C – 32°C), T_BASE (8°C – 18°C) and CAN_MX (0 mm – 10 mm). Moreover, we kept constant the discharge (HRU_SLP, SLSUBBN, CN_2) and groundwater (GW_REVAP, RCHRG_DP, REVAPMN) parameters. The parameters HRU_SLP and SLSUBBN are catchment specific and we kept the values which are derived in the model setup generation. The parameters CN_2, GW_REVAP, RCHRG_DP, and REVAPMN were transferred from studies in the Bétérou catchment (Duku et al., 2016) and from the SWAT-T analysis of Alemayehu et al. (2017).

With this new set of parameters, we ran all 24 benchmark elements again and analyzed the outcome. The conclusion is basically the same: LAI-AET modelling is inevitable for an adequate estimation of AET, particularly in the forested region. We adjusted the new version of the manuscript accordingly. Still, future work should address the implications of LAI-AET on the catchment scale. A multi-objective calibration of e.g., discharge, LAI, and AET should consider all relevant parameters and not only focus on LAI-AET. We are working on a study/publication with this regard currently where we analyze the Bétérou catchment in Benin with SWAT-T.

The manuscript is adjusted accordingly:
- Explanation for the constant parameters line 286-290
- Results in 3.3 Optimization and benchmark test of the LAI-AET modelling line 365

- There appears to be some confusion regarding the relationship between PET methods and LAI in the manuscript. Regardless of the PET method selection, PET is a model input derived from weather variables, while LAI is used only when computing potential plant transpiration. Given the availability of observed ET and LAI data in the study area, I highly encourage the authors to investigate the effect of PET method selection on ET simulations alongside a reliable representation of LAI. This would be a valuable addition to the manuscript, as PET method selection is often dictated by data availability not by its reliability for certain climatic regions.

Answer:
Thank you for questioning the relationship between PET and LAI. We tried to include the necessary equations and figures and explain why we chose all three methods.
The focus of the paper was the analysis of the role of LAI on AET with respect to measured data for both, LAI and AET. Considering the set of equations in Table 1 of the manuscript, LAI is a term in all three equations for the potential plant transpiration. Although PET itself is not directly dependent on LAI, the potential plant transpiration and hence AET is influenced by LAI. We argue that the different plant transpiration approaches (threshold based for PET-HG and PET-PT vs. more complex for PET-PM) are of relevance for the hydrological community to be investigated, particularly since LAI is a term in all three approaches and LAI-AET is the focus of this study.
Moreover, the actual plant growth (actual LAI) is determined with the plant growth factor, i.e., stress factors (Equation 5). The plant growth factor consists of water, temperature, nitrogen, and phosphorous stress where the water stress is directly linked to AET (Equation A1). Hence, a reciprocal interaction of LAI and AET is modelled in SWAT-T. This reciprocal interaction of LAI-AET can be observed in Figure 3 where the LAI parameters influence AET and vice versa.
Given this LAI-AET interaction, we wanted to focus on a holistic analysis where all relevant parameters but also all potentially relevant PET methods are included. For data-scarce regions like Western Africa, the information of which PET method to choice is relevant. We wanted to provide another piece of information for practitioners if the LAI-AET interaction is of objective.

Thank you for encouraging us to investigate the effects of PET method selection on AET. Actually, we are already working on a follow-up publication where we focus on the PET method choice and the LAI-AET interaction for the prediction of AET specifically. In a first step (with this study), we wanted to understand the LAI-AET interaction based on LAI which is relevant for the follow-up AET studies. Since measured data for AET is available, we plan to do a parameter ranking and a sensitivity analysis (Morris method). The work is in progress. Overall, we wanted to focus in the present study on LAI and its role for AET. With the presented in-depth analysis, we were able to show that we can adequately predict AET based on LAI only. We wanted to do the analysis on LAI in detail. We tried to avoid a manuscript which is loaded and which may not has the capacity to go into detail as we did for LAI.

- ET is a major component of the water balance in tropical climates. Therefore, presenting the simulated water balance components is relevant. However, for the reasons mentioned above, the grassland and forest could have similar water yield.

Answer:
We added the water balance components of each run to the appendix. Indeed, the water yield (surface runoff) differs for the land cover types.
Appendix A4, line 568

- The manuscript calibrates parameters relevant to biomass and net primary productivity, which are directly influenced by LAI. The authors should include in their recommendations an exploration of the influence of improved LAI simulation on biomass/primary productivity simulation, which is critical for water and carbon flux simulations of a watershed.

Answer:
For the evaluation of the computation of the biomass/primary productivity, different parameters in SWAT/SWAT-T are relevant. Yang & Zhang (2016), for example, investigated the biomass/primary productivity for flux sites of the AmeriFLUX network. They identified BIO_E, BLAI, T_OPT, T_BASE, and BIO_LEAF to be the most significant parameters for biomass. Yang & Zhang (2016) applied observed biomass data to particularly validate the modelled biomass. Apart from the BIO_LEAF parameter, we included the same parameters to model the actual plant growth (LAI). Our study focused to evaluate the LAI-AET interaction. Hence, we followed the suggestions of Alemayehu et al. (2017) and did not include the parameter BIO_LEAF in the LAI evaluation.
The parameter ranges for BIO_E and BLAI in our study are similar to the study of Yang & Zhang (2016). We kept the default BIO_LEAF parameter values which can be too high according to Yang & Zhang (2016). Since we are missing the data for biomass production, we can't give an evidence-based answer to the biomass production. Still, we assume that the biomass production based on a LAI calibration can give a good, first indicator of the magnitude of the biomass since LAI and biomass are linked. Still, in-depth analysis with observed biomass data is inevitable if the modelling objective is the evaluation of biomass and net primary productivity.

We adjusted the manuscript accordingly; lines 500-505.

- The manuscript also needs improvement in the writing. It is a bit too long for a case study presenting very specific cases.

We tried to shorten the manuscript and removed sections which are not concise, e.g., we shortened the Results, Discussion, and Outlook chapter. At the same time, we added few sentences since we wanted to improve the manuscript overall given the review comments. In addition, a mother-tongue English also did another proof-reading on the quality for the second version of the manuscript.

Please also see the track-changes file for the improvements in writing.

**Minor /edit comments**

P2 ln 26 rewrite the sentences "It varies notably dependent on land cover, soil.....". You may consider Its variability notably depends on ......

Answer:

We changed the manuscript accordingly; line 26

P2 ln38 improve the language "The usually small source area.....' You may consider this ' This is mainly attribute to the small foot print

Answer:

We changed the manuscript accordingly. Line 38

Figure 1. Remove the ' in 14000 samples

Answer:

We changed the figure accordingly. Figure 1, line 91

P3 ln65-68 "To the best of our knowledge, the reciprocal LAI-AET interaction and the relevance of a coupled LAI-AET parameter estimation have not yet been evaluated with respect to measured LAI and AET in SWAT/SWAT-T…". There are already a few published studies including

1. Abitew et al. 2023 . Innovative approach to prognostic plant growth modeling in SWAT+for forest and perennial vegetation in tropical and Sub-Tropical climates. Journal of Hydrology X
2. Haas et al. 2022. Improving the representation of forests in hydrological models. Science of the Total Environment

Answer:

Thank you very much for the literature. We integrated the work of both to our manuscript. Abitew et al. (2023) only used remotely-sensed LAI data from MODIS. Haas et al. (2022) didn't use measured AET data. Hence, the application of measured AET and LAI data has been used in our study for the first time to the best of ouar knowledge. We have reformulate the corresponding sentence and also included both suggested studies.

Changes: line 58, line 63 and line 68

P5 ln 105. Write "Tab.1 …' as Table 1 and edit similar instances throughout the manuscript

Answer:

We changed the manuscript accordingly.

P6 ln122 "Plant growth is modelled in SWAT/SWAT-T with the "Environmental Policy Impact Climate (EPIC) model (Arnold et al.,1998)…." Double check this and improve the language

Answer:

We improved the manuscript accordingly; line 124

P 9 ln 188-190. This sentence is confusing and needs improvement

Answer:

We reformulated the section.

P10 ln 205-2010 Use '-' or '/' to separate day-month-year

Answer:

We improved the manuscript accordingly.

P11 ln 250-251 Please add a text that briefly describes what Seibert et al. (2018) proposed for readers

Answer:

We improved the manuscript accordingly; line 256-261

P19 ln 393 '…figured a influences…' drop a

Answer: We improved the manuscript accordingly.

**List of most relevant changes**

Please note that we only list the most relevant changes. For changes concerning the minor comments, please refer to the answers we gave to the comments.

| Review comment | Change |
|---|---|
| Parameter choice and ranges | We simulated all 24 benchmark setups again with a new set of parameters. We defined constant parameter values for discharge, groundwater and geospatial parameters. The results of the new runs are similar. The respective changes are made in the manuscript. (line 286-290, Chapter Results) |
| Implication of improved LAI simulation on biomass/primary productivity | Since we are missing the data for biomass production, we can't give an evidence-based answer to the biomass production. Still, we assume that the biomass production based on a LAI calibration can give a good, first indicator of the magnitude of the biomass since LAI and biomass are linked. Still, in-depth analysis with observed biomass data is inevitable if the modelling objective is the evaluation of biomass and net primary productivity.

We adjusted the manuscript accordingly; lines 500-505. |

References:
Duku, C., Zwart, S. J., and Hein, L.: Modelling the forest and woodland-irrigation nexus in tropical Africa: A case study in Benin, Agriculture,

Ecosystems & Environment, 230, 105–115, https://doi.org/10.1016/j.agee.2016.06.001, 2016.
Yang, Q. and Zhang, X.: Improving SWAT for simulating water and carbon fluxes of forest ecosystems, The Science of the total environment, 569-570, 1478–1488, https://doi.org/10.1016/j.scitotenv.2016.06.238, 2016.

---

## Author Response (AR2)

**Author's response of Merk et al. (Hess-2024-131)**

A point-by-point response to the second Revision

**Point-by-point replay on Revision #2**

Dear Editor and Reviewers,

On behalf of all authors, I want to thank you again for your comprehensive feedback on our work. Your remarks are valuable for the publicationThe minor revision you suggested is the improvement of the English language in the manuscript. We handed our manuscript to a proofreading service at Technical University of Munich (https://www.gs.tum.de/fileadmin/w00bik/www/Attachments/Information_fuer_Promovierende/PDFs/20200505_Merkblatt_Lektoratsservice_barrierefrei.pdf last accessed: 10/24/2024).

The English language has been reviewed and improvements are made. The latest submission is proofread and, thus, it will improve its relevance for the community.

With kind regards
Fabian Merk